# Structure of a cleavage-independent HIV Env recapitulates the glycoprotein architecture of the native cleaved trimer

Anita Sarkar [1,2,3], Shridhar Bale[4], Anna-Janina Behrens[5], Sonu Kumar [1,2,3], Shailendra Kumar Sharma[1,2], Natalia de Val[1,3], Jesper Pallesen[2,3], Adriana Irimia[1,2,3], Devan C. Diwanji[3], Robyn L. Stanfield[1,2,3], Andrew B. Ward[1,2,3], Max Crispin[4,5,6], Richard T. Wyatt[1,2,4] & Ian A. Wilson [1,2,3,7]

Furin cleavage of the HIV envelope glycoprotein is an essential step for cell entry that enables formation of well-folded, native-like glycosylated trimers, releases constraints on the fusion peptide, and limits enzymatic processing of the N-glycan shield. Here, we show that a cleavage-independent, stabilized, soluble Env trimer mimic (BG505 NFL.664) exhibits a "closed-form", native-like, prefusion conformation akin to furin-cleaved Env trimers. The crystal structure of BG505 NFL.664 at 3.39 Å resolution with two potent bNAbs also identifies the full epitopes of PGV19 and PGT122 that target the receptor binding site and N332 supersite, respectively. Quantitative site-specific analysis of the glycan shield reveals that native-like glycan processing is maintained despite furin-independent maturation in the secretory pathway. Thus, cleavage-independent NFL Env trimers exhibit quaternary protein and carbohydrate structures similar to the native viral spike that further validate their potential as vaccine immunogen candidates.

[1] IAVI Neutralizing Antibody Center, The Scripps Research Institute, La Jolla, CA 92037, USA. [2] Center for HIV/AIDS Vaccine Immunology and Immunogen Discovery, The Scripps Research Institute, La Jolla, CA 92037, USA. [3] Department of Integrative Structural and Computational Biology, The Scripps Research Institute, La Jolla, CA 92037, USA. [4] Department of Immunology and Microbiology, The Scripps Research Institute, La Jolla, CA 92037, USA. [5] Oxford Glycobiology Institute, Department of Biochemistry, University of Oxford, Oxford OX1 3QU, UK. [6] Centre for Biological Sciences and Institute for Life Sciences, University of Southampton, Southampton SO17 1BJ, UK. [7] Skaggs Institute for Chemical Biology, The Scripps Research Institute, La Jolla, CA 92037, USA. Correspondence and requests for materials should be addressed to R.T.W. (email: wyatt@scripps.edu) or to I.A.W. (email: wilson@scripps.edu)

The HIV envelope (Env) glycoprotein trimer mediates viral entry into target host cells and is the only virally encoded surface antigen accessible to the humoral immune system on intact virions[1]. Most individuals with HIV develop strong autologous antibody responses against Env, but these antibodies are often unable to neutralize the wide diversity of circulating virus strains[2]. The primary goal of rational HIV vaccine development is to elicit broad and potent antibody responses against conserved Env epitopes found across diverse HIV clades and subtypes. Env-based immunogens that display the epitopes of broadly neutralizing antibodies (bNAbs) are crucial in the rational design of a universal HIV vaccine.

The trimeric precursor glycoprotein, gp160, is cleaved by cellular endoprotease furin resulting in the formation of non-covalently associated gp120/gp41 heterodimers. Furin cleavage liberates the fusion peptide (FP) at the N-terminus of gp41 and is required for infection of $CD4^+$ $CCR5^+$/$CXCR4^+$ susceptible host cells[3]. Historically, expression of soluble native Envs has been challenging, mostly due to the unstable nature of the non-covalently associated heterodimer. A soluble mimic of the furin-cleaved HIV Env, termed SOSIP, was designed to yield native-like trimers, and contained an engineered disulfide bond that covalently tethers gp120 to gp41 and an I559P mutation that favors the gp41 prefusion state and enhances Env trimerization[4, 5]. The SOSIP trimers are antigenically and structurally similar to wild-type Env[6–9].

HIV Env is heavily glycosylated with glycans contributing ~50% of its molecular weight[10]. The glycans act as a protective shield that almost completely covers the protein surface and helps evade immune surveillance. However, in the process, the dense patches of self N-linked glycans that are displayed on the Env surface become targets for the immune system as such extensive glycosylation is not found on host proteins. The glycosylation pattern on Env is also indicative of its folded state[11] as the compact soluble cleavage-dependent Env limits glycan processing to predominantly oligomannose, especially on gp120. Free gp120 monomers and uncleaved gp140s generally display more processed glycoforms[11–13] due to greater accessibility to the carbohydrate processing enzymes[14, 15].

The SOSIP design, like wild-type Env, requires furin cleavage to acquire a compact prefusion conformation, which adds complexity to vaccine candidate production, but which so far has been circumnavigated[16]. However, previous versions of uncleaved trimers were found to adopt non-native "open" conformations[17–19]. Thus, native flexibly linked (NFL) HIV Env trimers were designed to circumvent the requirement for cleavage by substituting the REKR cleavage site at the C-terminus of gp120 with a longer $2xG_4S$ peptide that covalently links gp120 to gp41[20]. The NFL design retains the trimerization-enhancing I559P mutation[5] from the SOSIP design, while dispensing with the disulfide bond[4] between gp120 $Ala_{501}$ to gp41 $Thr_{605}$[20].

Appropriate display of Env antigenic sites is a critical quality control measure for eliciting neutralizing antibodies, in addition to stability, homogeneity, and yield. Presently, six antigenic sites identified on HIV Env include the gp120 receptor (CD4) binding site (CD4bs), N332-centered V3 glycan supersite, V1–V2 loops at the trimer apex, gp120/gp41 interface, gp41 membrane-proximal external region (MPER), and the FP[21]. The dense glycan patch on the dominant N332 supersite is also involved in camouflaging the co-receptor binding site[22]. Central to antibodies that bind to this region is a structurally conserved epitope consisting of the $Man_9GlcNAc_2$ glycan (also called $Man_9$; N-acetylglucosamine is abbreviated as GlcNAc or NAG) on N332, and the G(D/N)IR sequence at the V3 loop base. These bNAbs vary in their usage of the surrounding glycans for binding and/or neutralization during the course of their evolution[23].

Constraints on glycan processing and flexibility are imparted by the quaternary arrangement of the trimer. The N332 supersite is relatively extensive, accessible and apparently the most immunogenic[24], and antibodies target this site via diverse angles of approach[22, 25]. By contrast, the CD4bs is recessed, discontinuous, highly conformational, and one of the most conserved sites on Env[26]. This epitope is targeted by the highly effective VRC01-class of antibodies[27]. The location of this critical site at the inter-domain interface, surrounded by the V1, V2, and V3 variable loops and multiple glycans, restricts antibody access around the CD4 binding loop[28]. Over the course of evolution during natural infection, bnAbs to this site acquire common features to negotiate the various obstacles that hinder binding to the CD4bs and that also enable additional quaternary contacts to be made within the context of the native Env trimer[7, 29, 30]. Although the VRC01-class of antibodies was thought to be difficult to elicit by vaccine immunogens owing to the bNAbs being highly somatically mutated and glycans obstructing antibody entry to the site, recent studies have demonstrated that appropriate bNAb precursors for targeting the CD4bs appear in sufficient frequency in the human antibody repertoire[31]. Hence, the CD4bs remains a high-value target for bNAb elicitation.

Here we present the crystal structure of BG505 NFL.664 in complex with bNAbs PGV19 (CD4bs) and PGT122 (N332 supersite) at 3.39 Å resolution. The cleavage-independent, soluble HIV Env design displays a compact prefusion structure and native-like glycosylation. These data further attest that the quaternary structure dictates the patterns of N-glycosylation and, therefore, the spectrum of glycoforms found at particular locations, on both SOSIP and NFL designs[6, 7, 11–13, 20, 29, 32]. The NFL structure provides direct comparison of the antigenic and glycosylation profiles of these two classes of Env-based vaccine candidates presently being tested in pre-clinical animal models prior to human testing.

## Results

**Structure determination of BG505 NFL.664.** The BG505 NFL.664 construct was expressed in HEK 293F cells supplemented with glycosidase inhibitors kifunensine and swainsonine and purified as previously described[33]. The purified BG505 NFL.664 trimer was incubated with 2× molar excess per binding site of Fabs PGT122 and PGV19 for stabilization, and further treated with EndoH glycosidase (New England Biolabs) that partially trims accessible glycans to the first GlcNAc moiety (Supplementary Fig. 1a-c). The BG505 NFL.664 trimer complex crystallized in the hexagonal $P6_3$ space group (Supplementary Table 1). Diffraction data were collected from the best diffracting crystal to 3.39 Å resolution with data completeness of 99.9% in the highest resolution shell (Supplementary Fig. 1d, e, and f). The structure was determined by molecular replacement (MR) using BG505 SOSIP.664 (PDB 4TVP) and unbound PGV19 Fab (determined during this study, Supplementary Table 1). The asymmetric unit contains one copy each of the BG505 NFL.664 protomer and Fabs PGT122 and PGV19 with 75.2% solvent (Fig. 1a). After rigid body refinement, good quality electron density was observed for BG505 NFL.664 Env and a substantial part of the N-glycan shield that was unaffected by EndoH treatment (78 saccharide moieties on 23 out of 27 N-linked glycosylation sites per protomer). The trimer complex was refined to $R_{cryst}$/ $R_{free}$ of 31.1%/ 32.6% and consists of Env residues $32_{gp120}$ to $664_{gp41}$ with 3 out of the 10-residue $2xG_4S$ linker being resolved; residues 149–150 and 186A-H in the V1–V2 loop (HXB2 numbering) are disordered. The FP (512–527) is well resolved even without a proximal antibody bound. PGT122 is stabilized by crystal contacts, while PGV19 makes no lattice

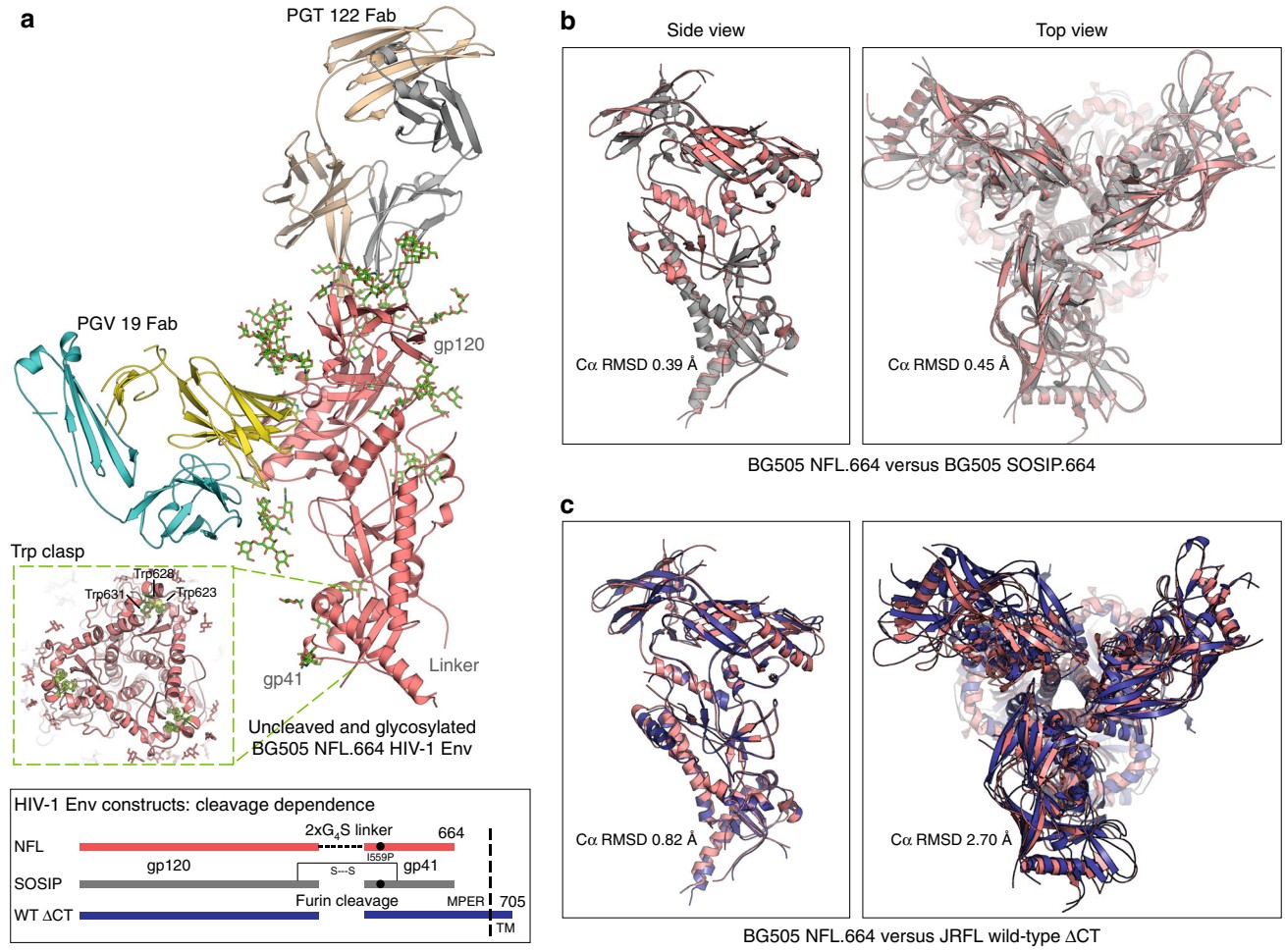

**Fig. 1** Molecular architecture and comparison of the soluble, cleavage-independent BG505 NFL.664 with natively cleaved designs. **a** The crystal asymmetric unit of the complex comprises one copy of the BG505 NFL.664 protomer (salmon) bound to one PGT122 Fab (LC: gray, HC: wheat) at the N332 site and PGV19 (LC: cyan, HC: yellow) at the CD4bs. Inset: The gp41 tryptophan clasps (green) are still present on the vertices of the NFL trimer's lower aperture. The bottom panel schematic illustrates the cleavage dependence and design variations between different trimeric Env constructs[16, 20, 37]. **b** Comparison of the crystal structures of BG505 NFL.664 and BG505 SOSIP (PDB 5CEZ, gray) (side view of protomer and top view of trimer). **c** Comparison of the crystal structure of BG505 NFL.664 and the cryo-EM structure of JRFL WTΔCT (PDB 5FUU, dark blue) (side view of protomer and top view of trimer). The large RMSD between the compared trimer structures arises from the asymmetry in the JRFL WTΔCT induced by binding of two PGT151 antibodies per trimer

interactions that results in its flexible constant region being disordered, as frequently observed in other Fab structures[34].

For electron microscopy studies, various CD4bs antibodies were tested for binding to BG505 NFL.664 to compare with BG505 SOSIP bound to PGV04[7]. VRC03 imparted the most stability to the complex (~2 °C) and had the highest affinity (Supplementary Fig. 2a, b). Thus, VRC03 was selected for single particle reconstruction by cryo-electron microscopy (cryo-EM). BG505 NFL.664 bound to VRC03 at 7.8 Å displayed a compact, native-like prefusion conformation (Supplementary Fig. 3), in agreement with the crystal structure (cross-correlation coefficient of 93%). Further comparison with the cryo-EM structure of BG505 SOSIP.664 bound to PGV04 Fab (EMD-5779)[7] illustrates the high similarity between the NFL crystal and the cryo-EM structures (Supplementary Fig. 3d).

**Architecture of BG505 NFL.664.** The BG505 NFL.664 clade A Env adopts a native-like, compact, prefusion conformation. The PGT122 Fab binds vertically, crowning the trimer at the

N332 supersite, while CD4bs Fab PGV19 binds laterally and more centrally around the trimer axis (Fig. 1a). The prefusion conformation of gp41 is preserved in the presence of the covalent flexible 2xG₄S linker and deletion of the natural REKR cleavage site. Four helices from gp41 embrace the two termini of gp120, being anchored by a tryptophan clasp involving W623, W628, and W631, and facilitated by M530 in the fusion peptide proximal region (FPPR), as observed in the cleaved SOSIP Env trimer[35]. The three Trp residues on each trimer lie at the vertices of the triangular opening at its membrane-proximal region, preserving the prefusion conformation of gp41, prior to major Env rearrangements that are initiated after receptor and co-receptor binding during the membrane fusion events for virus entry (Fig. 1a, inset). The cleavage-independent BG505 NFL.664 displays high similarity to previous crystal structures of the cleaved BG505 SOSIP.664 construct[6, 29, 30, 35, 36]. Among the available Env structures, BG505 NFL.664 differs in being a "single-chain format" i.e., with the furin cleavage site substituted by a 2xG₄S linker, and can also be differentiated from the BG505 SOSIP.664 design in lacking the 501C-605C disulfide bond (SOS), and from the wild-type, full-length native trimer in lacking

the MPER and cytoplasmic domains (ΔCT) (Fig. 1a, bottom panel, and Fig. 1b, c).

**Comparison with other clades and Env designs**. Comparison of BG505 NFL.664 and BG505 SOSIP structure (PDB 5CEZ[36]) shows Cα root mean square deviations (RMSDs) between the protomer/trimer to be 0.39 Å/0.45 Å, indicating that adding the NFL neither impacts the overall Env architecture nor its compactness (Fig. 1b). Superimposition of the clade A NFL crystal structure with the cleaved clade B JRFL native full-length ΔCT cryo-EM structure (PDB 5FUU[37]) produces protomer/trimer Cα RMSDs of 0.82 Å/2.70 Å, while RMSD values with the JRFL SOSIP.664 crystal structure (PDB 5FYK[38]) are 0.69 Å/0.78 Å. The larger RMSD from the JRFL WTΔCT trimer reflects some asymmetry when two PGT151 Fabs bind per trimer (Fig. 1c, Supplementary Fig. 4). Comparison of clade A BG505 NFL.664 with clade C 16055 NFL TD CC T569G (PDB 5UM8[39]) and clade G isolate X1193.c1 SOSIP.664 (PDB 5FYJ[38]) results in protomer/trimer Cα RMSDs of 0.7 Å/0.9 Å and 0.69 Å/0.76 Å, respectively, indicating that isolates from different clades with comparable stabilizing mutations do not vary much between each other (Supplementary Fig. 4).

**Unique features of BG505 NFL.664**. The structural features that distinguish BG505 NFL.664 from BG505 SOSIP.664 are the HR1 N-terminal (HR1_N) region (residues 548–568) of gp41, a fully resolved FP, the linker connecting gp120/gp41 that eliminates the furin cleavage site, and the absence of the 501C-605C disulfide bond (SOS) (Fig. 2a).

The helical propensity of HR1_N of BG505 NFL.664 is disrupted only by the I559P trimer-stabilizing mutation (Fig. 2b), representing a more ordered structural intermediate conformation for this flexible region than previously observed, although it appears to be sequence dependent based on multiple EM reconstructions (Supplementary Fig. 5). The JRFL WTΔCT cryo-EM structure, that lacks any stabilizing mutations including the I559P mutation, has a helical HR1_N (Supplementary Fig. 5a). The flexible linker in BG505 NFL.664 might allow more conformational flexibility around the gp120/gp41 interface allowing HR1_N to adopt a partial helical conformation consisting of two short helical segments disrupted by I559P.

We observe for the first time the entire FP of the BG505 isolate in the absence of interacting bnAbs, such as VRC34.01[40], ACS202[41], or PGT151[37]. The FP (residues 512–527) on BG505 NFL.664 forms a loop that is similar, but not identical, to the VRC34.01-bound state on BG505 SOSIP[40] (Fig. 2c, upper inset).

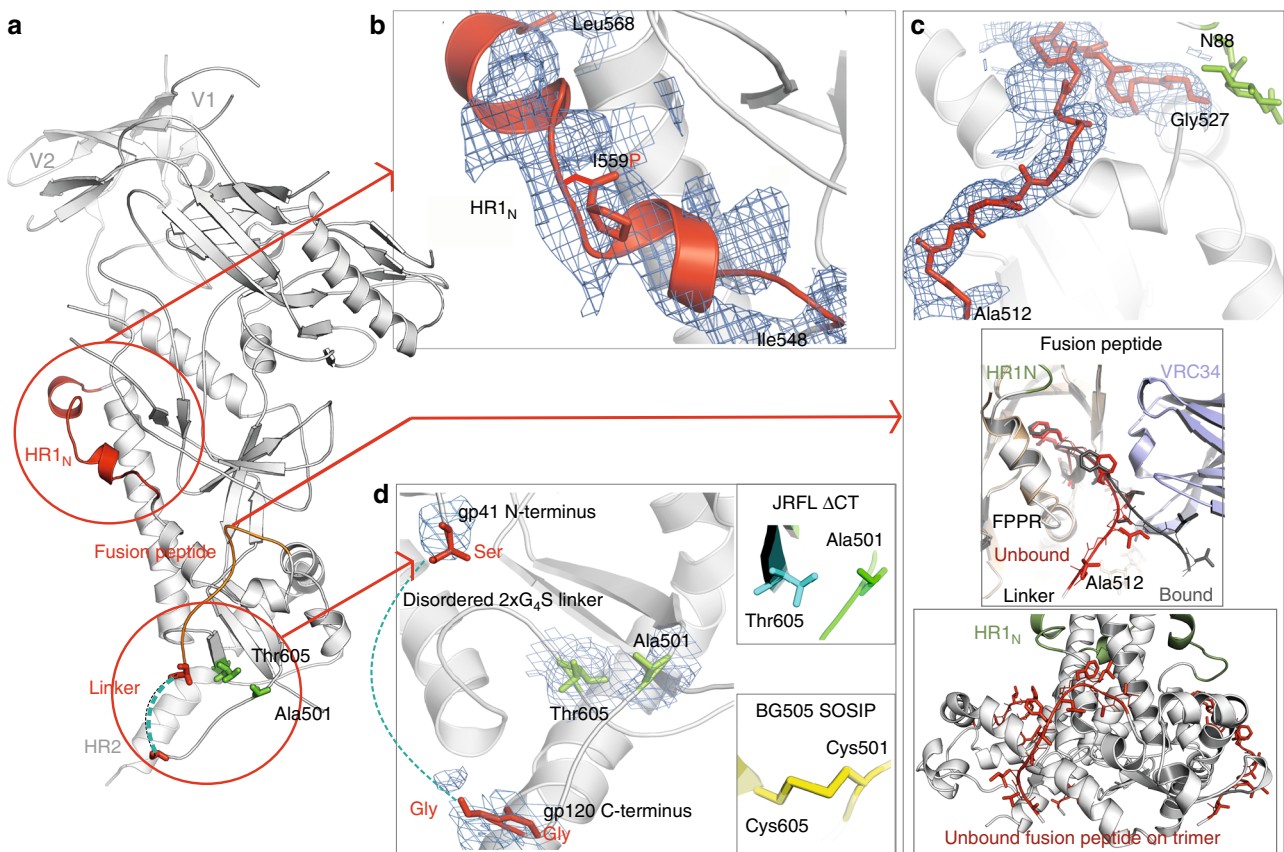

**Fig. 2** Distinguishing features of BG505 NFL.664. **a** Features of BG505 NFL.664 that do not appear in the cleaved BG505 SOSIP.664 structure are highlighted. Most of the flexible linker (dark green) is disordered in the crystal structure. The NFL design lacks the disulfide bond (SOS) between residues 501 and 605, and retains the original Ala and Thr at the respective positions in the primary sequence. **b** The 20-residue HR1_N region forms a helix (red cartoon) with the I559P trimer-stabilizing mutation disrupting it. **c** The backbone atoms of the FP (residues Ala512 to Gly527) are shown as observed in the BG505 NFL.664 structure. Top inset: FP conformations in the unbound (red on BG505 NFL.664) and bound (dark gray on BG505 SOSIP.664) to VRC34.01 (slate blue, PDB 5I8H) on the BG505 NFL.664 (white) and BG505 SOSIP.664 (wheat), respectively. The HR1_N (green), FPPR, and the flexible linker are labeled for context. Bottom inset: The unbound FP (red) is shown on the BG505 NFL.664 trimer. **d** Three residues (red) of the 2xG_4S linker are observed in the crystal structure. The insets compare the residue conformations at positions 501 and 605 in JRFL WTΔCT (lacking the SOS mutation like NFL) and SOSIP designs. The $2mF_o-F_c$ electron density composite omit maps (blue mesh) surrounding these features are contoured at 1.0σ

Nine FP residues (519–527) in BG505 NFL.664 are flanked by the N88 glycan, as in the VRC34.01-bound SOSIP, and stabilized by the FPPR and the symmetry mate of the heavy chain constant region of PGT122. The remaining FP residues (512–518) in BG505 NFL.664 turn back toward the gp120 C-terminus when attached to the flexible linker while, in VRC34.01-bound SOSIP, these extend toward the CDR loops; such a difference may affect antibody binding in this region due to the unavailability of a free and charged FP N-terminus. The unbound FP on BG505 NFL.664 is solvent exposed (Fig. 2c, lower inset) and its conformation is distinct from the PGT151-bound JRFL WTΔCT conformation.

BG505 NFL.664 resembles JRFL WTΔCT in lacking the disulfide between gp120 Ala$^{501}$ and gp41 Thr$^{605}$, with similar conformations of these residues, distinguishing it from the SOSIP design (Fig. 2d and insets). About 80% of the 2xG$_4$S linker in the NFL construct is composed of glycine residues that account for its inherent flexibility. Nevertheless, three residues at the linker's N- and C-termini were ordered in the crystal structure (Fig. 2d), but not observed in the cryo-EM map (Supplementary Fig. 3c).

**Mapping linker flexibility and inherent movements.** To ascertain the range of movement spanned by the 2xG$_4$S linker, we performed molecular dynamics (MD) simulations and analyzed the 250 ns trajectory. The glycosylated gp140 structure stabilizes after 50 ns of simulation in explicit solvent medium (Fig. 3a). The 2xG$_4$S linker is very flexible, as expected, with a root mean square fluctuation (RMSF) of ~7.5 Å compared to residues in its vicinity (Fig. 3b, Supplementary Movie 1). We analyzed the MD trajectory as a function of time for information on protein dynamics and conformational stability, starting from the conformation captured in the crystal structure. In ascending order of flexibility are the CD4bs < V5 < V1 < V2 < V4 < V3 < HR1$_N$ (specifically

residues 553–564) < 2xG$_4$S linker (Fig. 3b, c). The substantial motion of the linker explains its disorder in cryo-EM and crystal structures.

**Structural insights from BG505 NFL.664 comparisons.** A relative shift between the gp120/gp41 domains on the trimer was observed after aligning BG505 NFL.664 and other published Env structures (Fig. 4a). Such a rotation between gp120/gp41 subdomains is also observed between the stalk and head regions of influenza hemagglutinin in different subtypes[42, 43]. The gp120/gp41 rotations between multiple structures of the same HIV isolate or those belonging to separate clades are within the range of those observed in the hemagglutinin.

The base of the spike, comprising three gp41 domains, also tends to deviate more than the gp120s in all of the other structures irrespective of the epitope(s) bound by antibodies. These differences are shared between clades as well as different Env trimer designs highlighting some inherent flexibility associated with the Env spike. Other intra-domain conformational changes on gp120 have been described, where different populations are associated with repositioning of the V1 loop with respect to either V4 or V5 loops of the outer domain[44]. We already know that HR1$_N$ is a highly flexible loop in the SOSIP structures. Aligning the more-structured helical HR2 (residues 520–664) of BG505 NFL.664 (red) with other Env structures (various colors) reflects the variation in this region of gp41 (Fig. 4a, b).

To quantify the structural effect of combining antibodies on Env compactness versus the unliganded HIV trimer, we measured Cα distances (Å) at trimer apex (R166 on V2 loop) and at the base (E654 on HR2) in the available Env designs and compared them to the BG505 NFL.664. The general trend indicates that antibodies binding to one or more Env epitopes have significant,

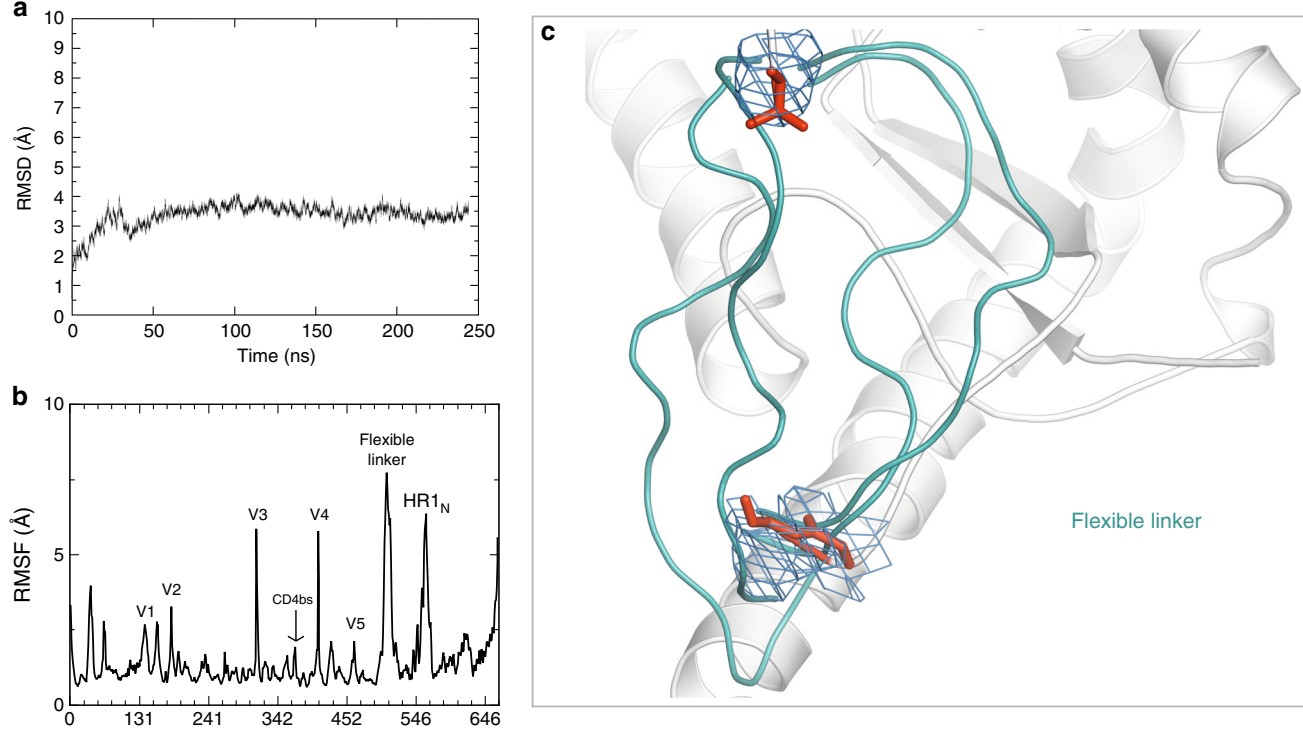

**Fig. 3** Linker flexibility and inherent motion in BG505 NFL.664. **a** Stabilization of the MD trajectory of BG505 gp140 NFL.664 simulated for 250 ns. **b** Root mean square fluctuations of the variable loops of BG505 NFL.664 over the 250 ns trajectory. **c** The range of movement of the 2xG$_4$S linker sampled over the MD simulation trajectory

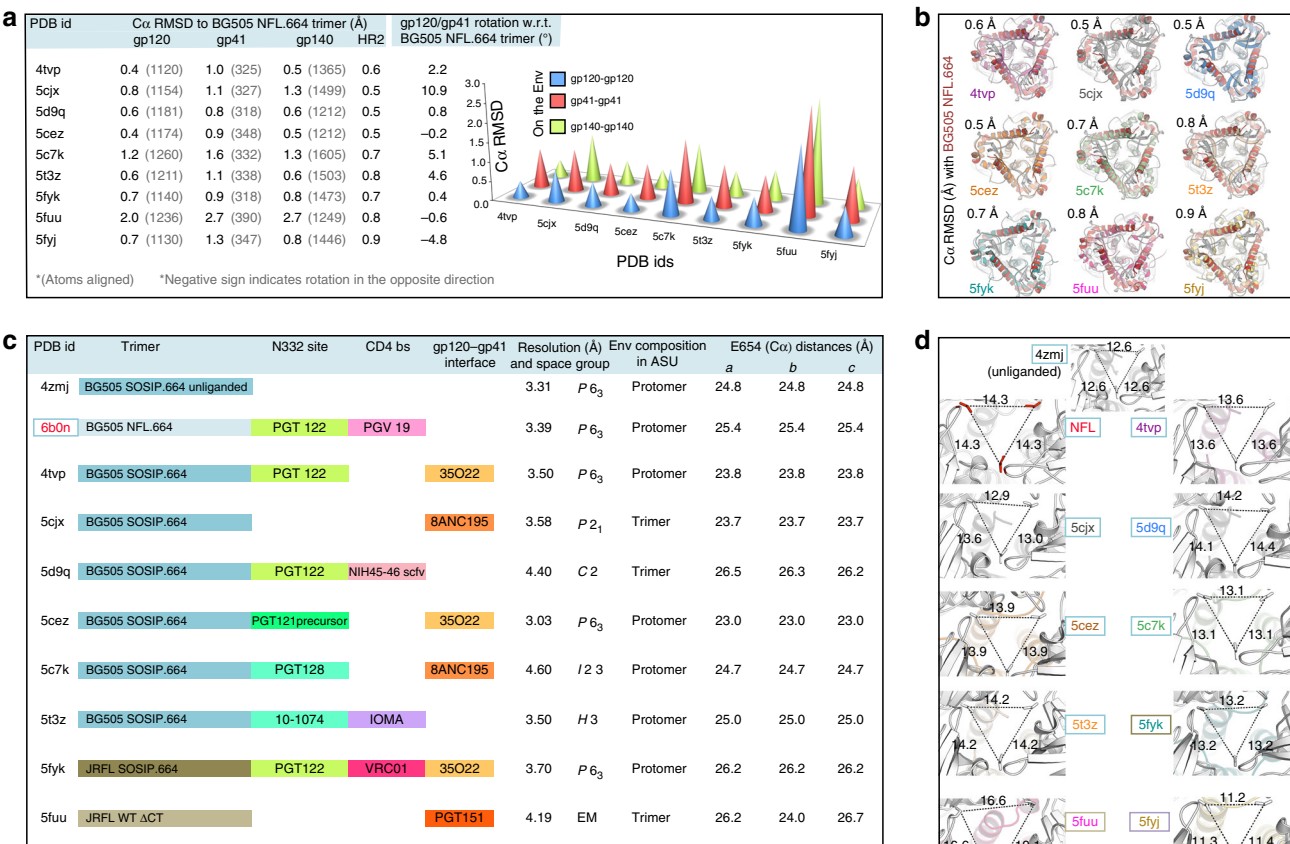

Fig. 4 Inherent flexibility and effect of the binding of antibodies on trimer opening. **a** Inter-domain flexibility observed by aligning the trimeric gp120 and gp41 individually on the Env spike for available HIV trimer structures, in contrast to the overall Cα RMSDs (Å) of the trimeric gp140s also illustrated on the right. **b** Contribution of HR2 to the movement observed at the base of the HIV Env constructs measured in Cα RMSD (Å) as in **a**. Rotation (°) between the gp120/gp41 domains measured on the soluble Env with respect to BG505 NFL.664. To calculate subdomain rotation, the angle between gp120/gp41 was measured using three points on the Env, namely, Cα atoms of the conserved W571 and Q591 on HR2 and a pseudoatom placed at the center of mass (https://pymolwiki.org/index.php/Center_of_mass) of gp120. This value was then subtracted from the angle measured for BG505 NFL.664 to infer rotation between the subdomains. **c** A list of the available Env trimer structures with a description of the epitope(s) occupied, resolution (Å), space group, Env component in the asymmetric unit (ASU), and the distances between E654 on each protomer marking the coordinates at the Env base as sides of the triangle a, b and c. **d** Inter-V2 distances (Å) measured between R166 (Cα) on each gp41 at the trimer apex for available HIV trimer structures of various isolates and designs in **c**

but possibly uncorrelated effects, on gp41 compactness (~3 Å) (Fig. 4c: sides a, b and c of Env base), and trimer apex opening (≥1 Å) when comparing different Env clades and designs to the unliganded structure (Fig. 4d). The BG505 SOSIP and NFL, with N332 and CD4bs sites both occupied, demonstrate identical apex opening (NFL and PDB 5T3Z[29] and 5D9Q[30]).

The compactness of Env apex also varies to some extent between isolates of different clades bound to the same antibodies (PDB 5FYK and 5FYJ[32]), when comparing JRFL SOSIP.664 (clade B) and X1193.c1 SOSIP.664 (clade G). Thus, the inherent flexibility of Env, combination of epitopes occupied, Env immunogen design, and the clade under study, all influence the compactness of the trimeric immunogen.

## Glycosylation of the cleavage-independent BG505 NFL.664.

Clear electron density for 23 out of the 27 potential glycosylation sites (~85%) on the BG505 soluble Env protomer was observed ($2F_o−F_c$ at 0.8–1.0σ) in the BG505 NFL.664 crystal structure (Figs. 5a and 6a). Density for the remaining N-linked GlcNAc residues was observed at lower sigma levels of the electron density map at N398 ($2F_o−F_c$ at 0.5σ), N411 and N462 ($2F_o−F_c$ at 0.7σ),

while the protein backbone for N405 was disordered (Fig. 6a); thus, these four glycosylation sites were not modeled into the structure or considered for further analysis. When compared to fully glycosylated BG505 SOSIP.664 (PDB 5FYL[38]) expressed in HEK 293T GnTI$^{−/−}$ for which ~3.8 saccharide moieties/sequon were observed on the structure, we found ~3.4 saccharide moieties/sequon on BG505 NFL.664 even with EndoH treatment after complex formation with PGV19 and PGT122. Beside the glycans at N332 and N276, which are expected to be shielded from glycosidase activity in this particular complex, many other glycans are protected by PGT122 and PGV19. Some of the larger glycans observed on the trimer surface (Fig. 5a) are clustered on the glycan shield and stabilize each other by inter-glycan interactions, such as N276-N234, N262-N448, N386-N363-N392, similar to EndoH-untreated clade G X1193.c1 SOSIP.664 and clade B JRFL WTΔCT structures (JRFL lacks N234), but not previously described in BG505 constructs with various antibodies bound, as listed in Fig. 4c. In the crystal structures, not all of the glycans or glycan moieties that are present on the trimer can be seen due to conformational flexibility. Individual glycans observed at N332 ($Man_9$), N156 ($Man_5$), and N301 ($Man_2$) are contacted by PGT122, consistent with the observation that PGT122 protects

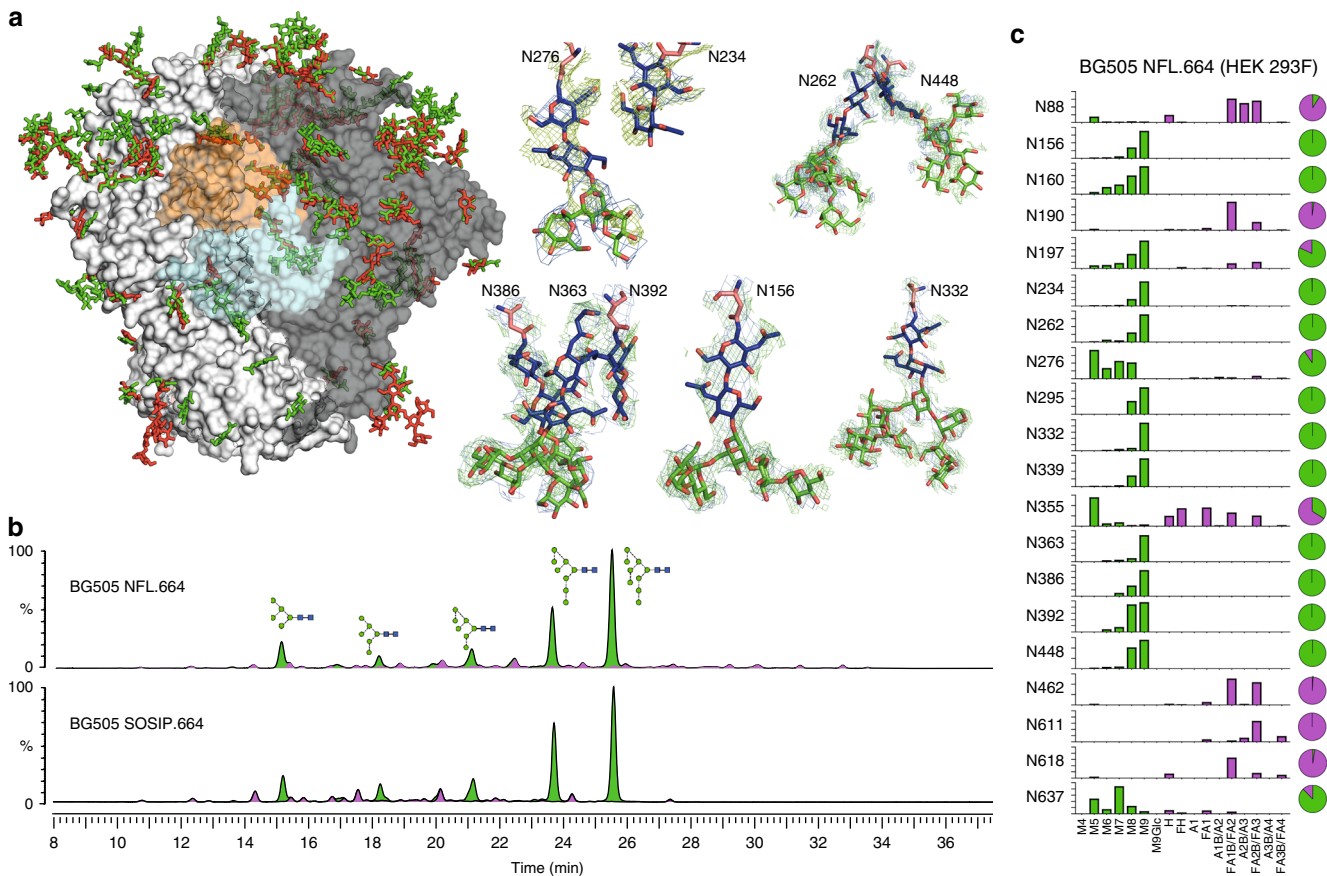

**Fig. 5** Glycosylation on the uncleaved, soluble NFL Env trimer. **a** On the left, superimposition of oligomannose glycans (green) observed on the BG505 NFL.664 trimer (shades of gray) versus the non-deglycosylated BG505 SOSIP.664 (red; PDB 5FYL) illustrates the protection by the bound antibodies from EndoH trimming, and the glycan fence around the PGV19 footprint (LC$_{var}$: cyan, HC$_{var}$: orange). On the right are various conformations of well-defined glycan sub- or super-structures observed in the BG505 NFL.664 crystal structure. **b** HILIC-UPLC spectra of fluorescently labeled N-linked glycans released from BG505 NFL.664 and BG505 SOSIP.664 produced in HEK 293F cells. Oligomannose and hybrid glycans are colored in green and complex glycans in pink. The corresponding structures for the oligomannose glycans (Man$_{5-9}$GlcNAc$_2$) are shown in the upper panel. **c** Site-specific N-glycosylation analysis of BG505 NFL.664. Relative quantification of the micro-heterogeneity of 20 out of the 27 N-glycosylation sites of BG505 NFL.664 produced in HEK 293F cells were determined, with the color scheme preserved from panel **b**. Glycoform annotation: oligomannose glycans Man$_5$GlcNAc$_2$ to Man$_9$GlcNAc$_2$ are represented as M5 to M9, hybrids as H, and fucosylated hybrids as FH. Complex glycans are also categorized by the number of branching antennae (A$n$); number of galactose residues (G$n$); and core fucose (f)

the glycans at N332 and N156 from EndoH access[6]. But surprisingly, despite being important for recognition by the PGT121-family of bNAbs[23, 36], glycans at N137 (GlcNAc) and N301 (Man$_2$) are still largely disordered, possibly indicating that specific interactions occur only with the chitobiose core for the PGT121-bNAb family. The N332 glycan rises ~20 Å from the Env surface and, along with the surrounding glycans N137, N156, and N301 that are important for this family of Abs, shields the underlying viral protein surface.

It has been previously reported that glycosylation in cleaved Env trimers is characteristically different from uncleaved trimers, particularly as foldon constructs, with the latter displaying more elevated glycan processing resulting in more complex glycans[11, 12]. The differences in glycan processing have been attributed to non-native open conformations of the uncleaved Env trimers providing access to glycosidases for further processing of oligomannose glycoforms[11, 32]. To assess the differences in glycosylation between BG505 NFL.664 and BG505 SOSIP.664, we recorded hydrophilic interaction liquid chromatography-ultra performance liquid chromatography (HILIC-UPLC) spectra of fluorescently labeled N-linked glycans released for both Envs, following identical purification steps after

expression in HEK 293F cells. Their overall glycosylation profiles were found to be highly similar with predominantly oligomannose glycoforms (Fig. 5b). An extensive site-specific glycan analysis (Fig. 5c and Supplementary Data 1 and 2) found the glycosylation profile at each site on BG505 NFL.664 to be comparable to that observed on BG505 SOSIP.664[13], which was purified using PGT145, except at N160 where SOSIP contains mixed populations of oligomannose and complex glycans, while NFL was found to contain exclusively oligomannose glycoforms here. N160, along with N156, N197, N262, N276, N301, and N637, on the apex and protomer interfaces, form the trimer-associated mannose patch of the BG505 isolate[12]. N160 lies at the trimer apex and may have slightly increased glycan processing, possibly due to trimer breathing resulting in minor variations in glycan processing, which is reflected in minor complex glycan populations observed for BG505 SOSIP[12]. In particular, we infer from these results that conserved oligomannose micro-clusters[13] that wrap around the outer domain of Env are indeed an important feature of compact, prefusion state, soluble Env designs.

The overall pattern of glycosylation and site-specific glycan composition observed in the cleaved (SOSIP) and cleavage-

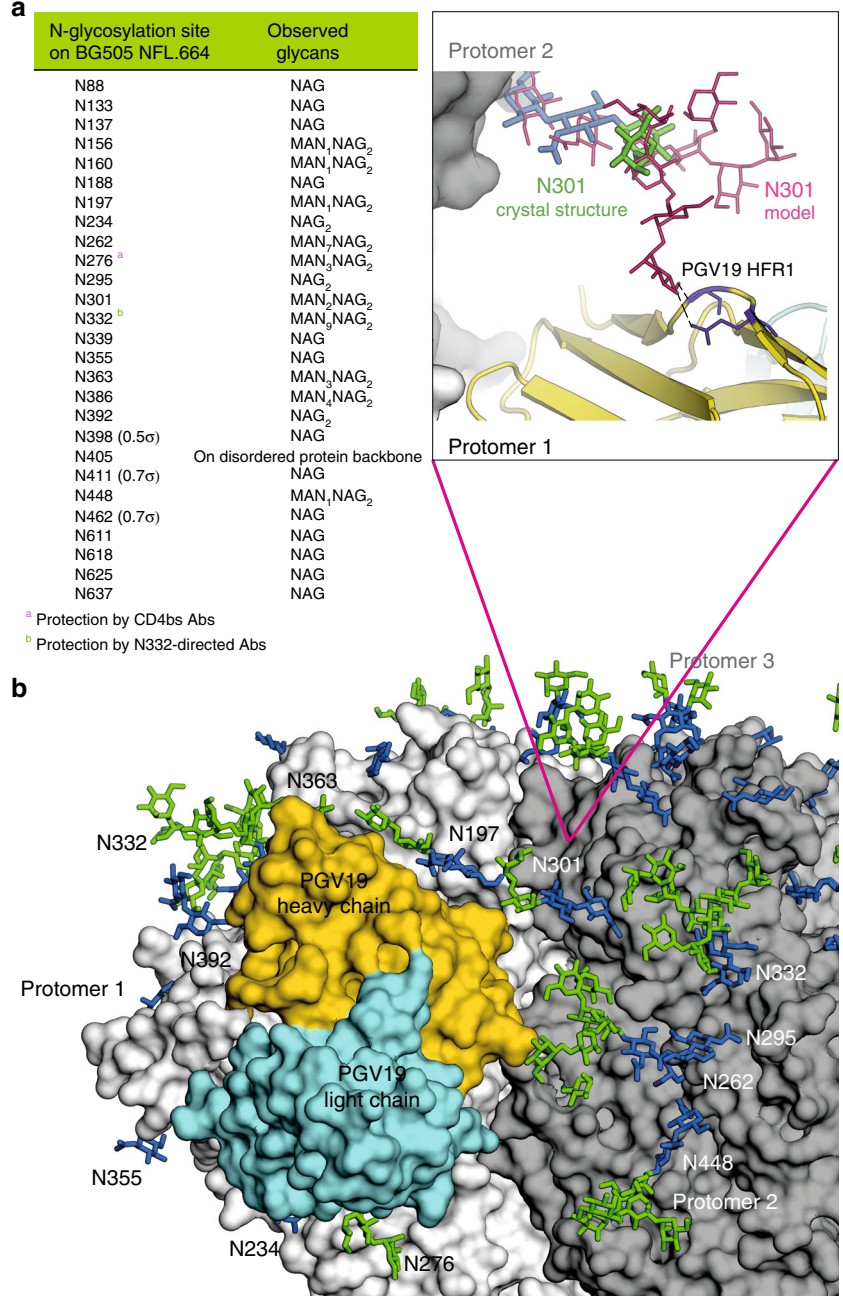

**Fig. 6** Glycosylation on the NFL Env trimer and glycan fence around the CD4bs. **a** Glycosylation observed in the BG505 NFL.664 structure (with at least 0.8–1.0σ electron density). Glycans that are observed at less than 0.8σ have not been included for calculating observed moieties per sequon or modeled on the structure, but are included in parentheses in **a**. **b** Surface representation of BG505 NFL.664 (protomers 1–3 shown in shades of gray) with the PGV19 variable region (HC: yellow, LC: cyan) and the N-glycosylation observed in the crystal structure (*N*-acetyl-glucosamine in blue and mannose in green). The glycan fence is formed by N197, N363, N234, and N276 on the same protomer, along with N301, N295, and N262 on the adjacent protomer of the Env trimer that the CD4bs Abs have to circumnavigate to access their epitope. The zoomed-in panel illustrates the modeled interaction of Man$_9$ at N301 with the heavy chain framework 1 (HFR1) of PGV19

independent (NFL) varieties converge, even though they may have associated differences in folding kinetics during glycoprotein secretion[45–47]. The presence of an optimal linker of sufficient length and flexibility in the single-chain NFLs possibly leads to adoption of a compact form shortly after protein folding in the ER and consequently prevents the erosion of the trimer-associated mannose patch observed previously in uncleaved HIV Env designs lacking the furin-cleavage site[11, 12, 48].

**The glycoprotein epitope of the PGV19 antibody.** The CD4bs is a difficult epitope to access as it is surrounded by a glycan fence (including N276 in the CD4bs Ab footprint area) that has to be bypassed or accommodated by an incoming antibody. All VRC01-class HIV antibodies isolated to date against the CD4bs from HIV-naïve[31] as well as infected persons are kappa chain antibodies, except for the sub-group of PGV19, PGV19b, PGV20, and PGV20b, which are lambda chain antibodies (all isolated from the same donor[27]). However, this is the first report of the

interaction of a VRC01-class lambda antibody in the context of the HIV Env trimer (Fig. 7a). The selection or possible advantage of kappa over lambda chain antibodies targeting the CD4bs within the potent VRC01-class is yet unknown, but is in contrast to N332-directed Abs which are largely lambda. The unbound PGV19 structure (Supplementary Fig. 6a, b) reveals that it is pre-configured for binding to the CD4bs, including the short CDRL3 to accommodate N276 (Supplementary Fig. 6c). PGV19, as well as VRC03 (used here in the EM characterization of BG505 NFL.664), have the signature features of VRC01-class of antibodies[49] (Supplementary Fig. 7). Although PGV19 and VRC03 are derived from the same heavy chain V-gene (IGHV1-2*02) like VRC01 and the naïve human antibodies previously characterized[31], they developed from different light chain precursors: PGV19 from IGLV2-14, VRC03 from IGKV3-20[27], and the naïve VRC01-HuGL2 from IGKV4-1[31], and variations in CDR length and/or conformation are observed in L1, L2, and H3 (Fig. 7b, c). While a 5-residue CDRL3 is conserved for accommodating the N276 glycan in all VRC01-class Abs (Supplementary Fig. 7a), both PGV19 and VRC03 show subtle differences in the signature VRC01-class interactions at the CD4bs. For instance, different

rotamers of $R71_{HC}$ and $E96_{LC}$ in PGV19 permit only a single hydrogen bond between gp120 D368 and N280 when compared to VRC01 (PDB 3NGB) and VRC03 (PDB 3SE8) (Supplementary Fig. 7b, d). Another hydrogen bond between CDRH3 W100a (Kabat numbering) and $N279_{gp120}$ that is found in PGV19 (W100b in VRC01) is absent in VRC03 due to F100d instead of a Trp at this position (Supplementary Fig. 7c). PGV19 and VRC03 both have an $N58S_{HC}$ mutation, thus lacking the hydrogen bond formed with the backbone of gp120 R456 by VRC01 (Supplementary Fig. 7e). PGV19 also lacks the hydrogen bond between CDRH2 W50 and gp120 N280 that exists in the Env complexes with VRC01 and VRC03[28, 50]. PGV20[27] (PDB 4LSU), another lambda-class antibody isolated from the same donor, has highly similar interactions to PGV19 except for F91 (like VRC03) instead of Y91 on the tip of its CDRL3 and $N58_{HC}$ (like VRC01) instead of S58 (as found on PGV19 and VRC03).

The NFL crystal structure illustrates the extensive glycan shield and interactions of PGV19 with the surrounding glycan fence with N363 on top, N234 and N276 below, and N197 from the same protomer, as well as $Man_9$ modeled at N301, and N295 from the adjacent protomer (Fig. 6a, b). The glycans at N197 and

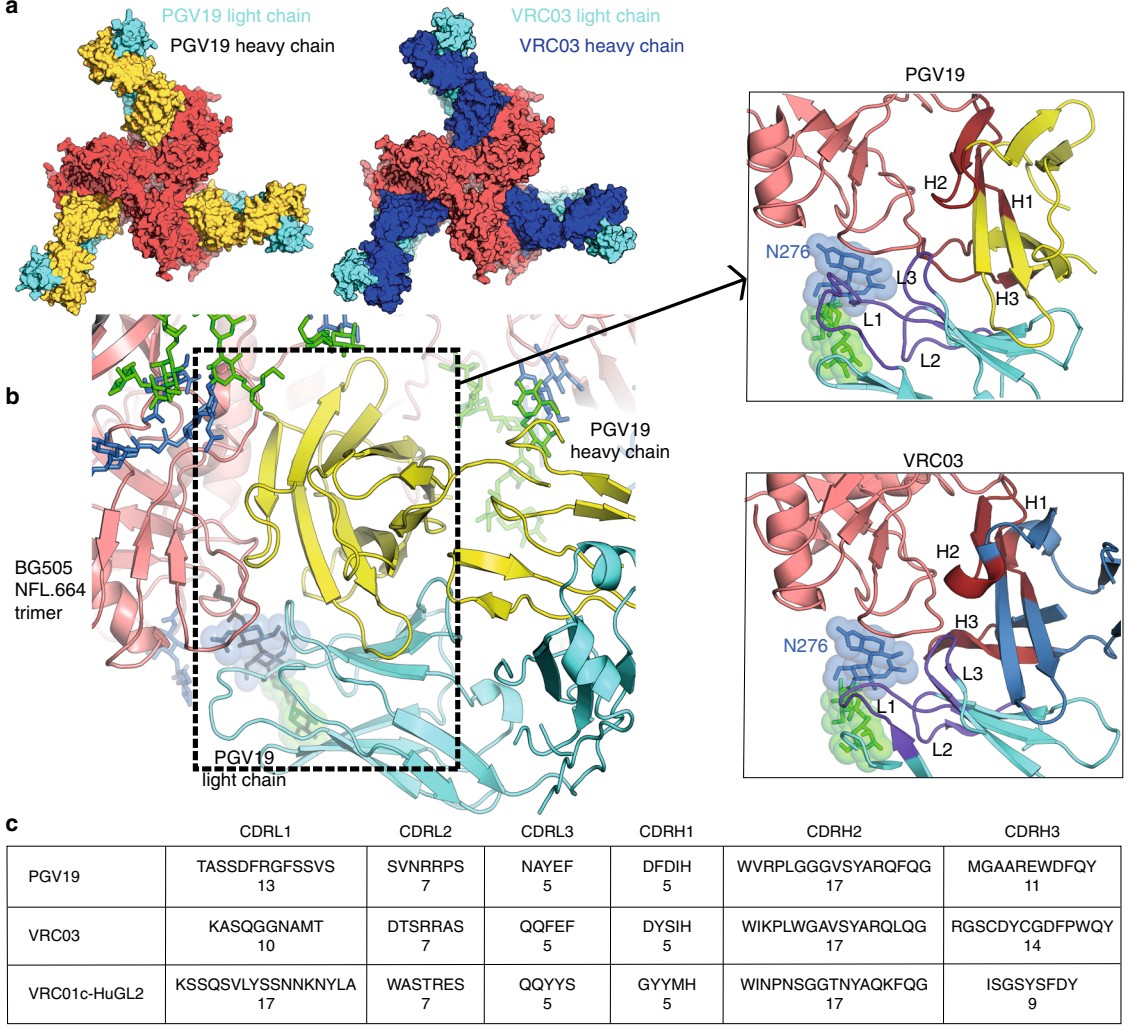

**Fig. 7** Similarities and differences between VRC01-class lambda and kappa antibodies. **a** Top view in surface representation of the BG505 NFL.664 trimer in complex with PGV19 (crystal structure; HC: yellow, LC: cyan) and VRC03 (model based on EM map; HC: dark blue, LC: cyan) used in this study for crystallography and cryoEM experiments, respectively. **b** CD4bs interactions of PGV19 with BG505 NFL.664 as observed in the crystal structure. Inset: The lambda (PGV19) and kappa (VRC03) light chains both evolved a short CDRL3 to accommodate the N276 glycan, and CDRH1 is less than 11 residues to prevent clashes with the $V5_{gp120}$ loop in Env (CDRHs: brown, CDRLs: purple). **c** Comparison of the CDR loop lengths of PGV19 and VRC03[50] from HIV[+] elite neutralizers and a naïve VRC01c-HuGL2[31] from an HIV-uninfected person

| | CDRL1 | CDRL2 | CDRL3 | CDRH1 | CDRH2 | CDRH3 |
|---|---|---|---|---|---|---|
| PGV19 | TASSDFRGFSSVS 13 | SVNRRPS 7 | NAYEF 5 | DFDIH 5 | WVRPLGGGVSYARQFQG 17 | MGAAREWDFQY 11 |
| VRC03 | KASQGGNAMT 10 | DTSRRAS 7 | QQFEF 5 | DYSIH 5 | WIKPLWGAVSYARQLQG 17 | RGSCDYCGDFPWQY 14 |
| VRC01c-HuGL2 | KSSQSVLYSSNNKNYLA 17 | WASTRES 7 | QQYYS 5 | GYYMH 5 | WINPNSGGTNYAQKFQG 17 | ISGSYSFDY 9 |

N276 show heterogeneity in composition[13, 29], thus having a capacity to mask an area with a radius of ~20 to 35 Å (by oligomannose and sialylated complex N-glycans, respectively) of the underlying protein surface around each of these glycosylation sites. Removal of one or more glycosylation site from the Env glycan fence around the CD4bs will increase exposure of the underlying protein surface to the immune system. This might then facilitate the initial engagement of germline antibodies[51, 52] to guide them toward evolving into bNAbs (although N276 requires re-introduction at a later stage of the vaccine regimen[51]). Thus, PGV19 makes multiple stabilizing contacts with its glycoprotein epitope and the surrounding glycan fence in the interprotomer setting of the CD4bs observed on the HIV Env trimer that it lacks on smaller Env immunogens. Also, such extensive interactions between N-glycan antennae of the N332 supersite and CD4bs-directed antibodies indicate a dependence of the latter on the surrounding glycan fence and the glycosylation profile of the Env trimer during maturation (Fig. 6b, inset).

**Glycans allosterically modulate binding at Env epitopes.** In the BG505 NFL.664 complex structure, although PGT122 and PGV19 are located distally on the same gp140 protomer (Fig. 8, top illustration), they are in proximity on adjacent protomers of the trimer (Fig. 8, middle illustration), with their variable regions being ~23 Å apart (Fig. 8, bottom illustration). We investigated if this proximity translated to an allosteric and/or steric effect in sequential binding of the two antibodies, and also the role of N276, which has to be accommodated by all known CD4bs-targeting antibodies.

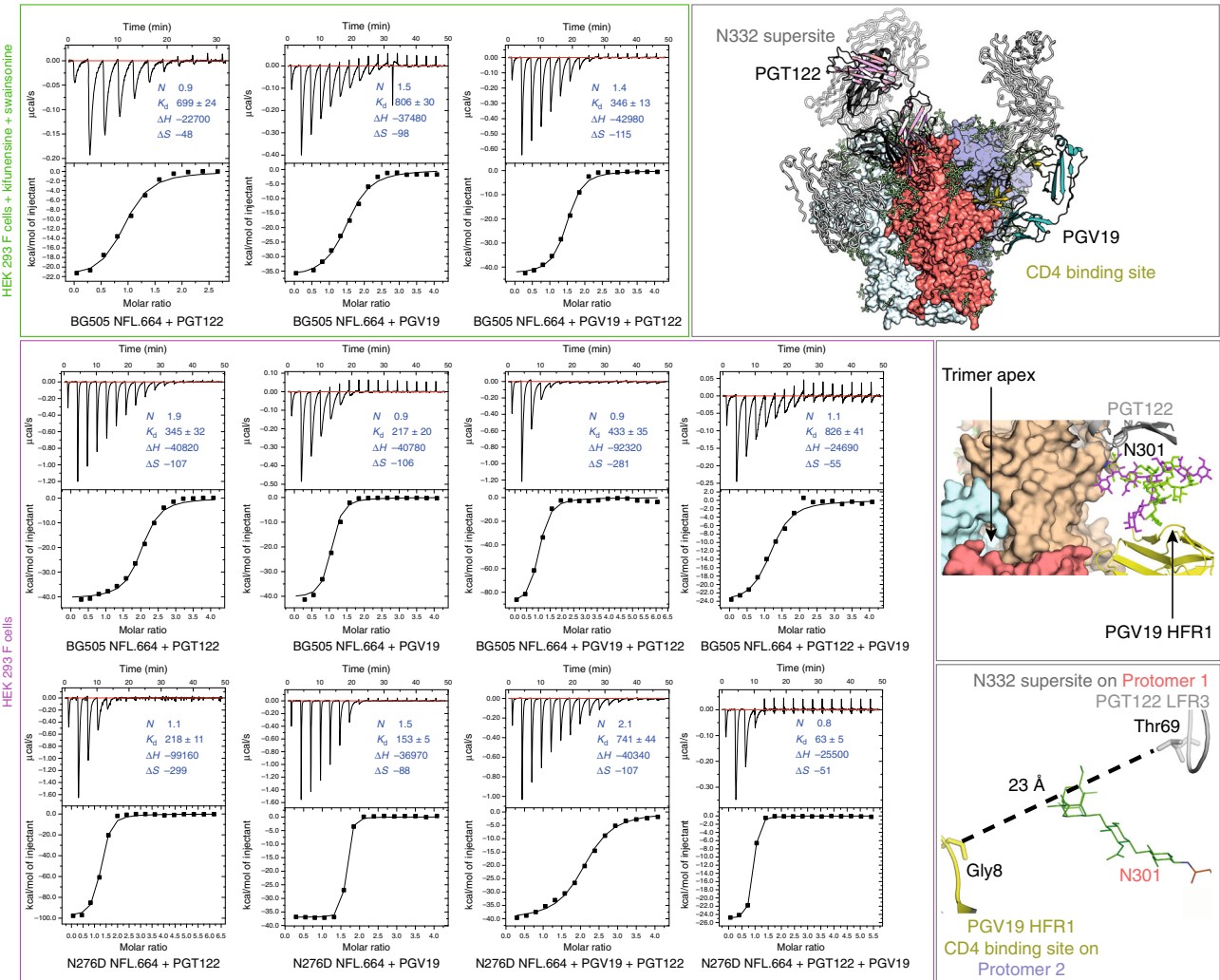

**Fig. 8** Biophysical characterization of sequential binding events of bnAbs with the NFL Env trimer. The top panel illustrates binding by ITC of PGT122 and PGV19 individually and then sequentially to BG505 NFL.664 produced in HEK 293F cells to which kifunensine and swainsonine were added (resulting in oligomannose glycoforms only) and depicted within the green outlined box). The top right illustration highlights the location of PGT122 and PGV19 at the N332-supersite and the CD4bs, respectively, on one protomer as a colored cartoon. The three protomers of the Env trimer are shown in separate colors in surface representation. The second panel shows the binding of the individual antibodies and then their successive binding to BG505 NFL.664 produced in HEK293F cells (comprising of oligomannose, complex and hybrid glycans as detected by mass spectrometry studies, outlined by a pink box). The middle illustration on the right shows different glycoforms modeled on N301 (Man9: green, complex glycan: magenta) and the location of PGV19 and PGT122 as viewed from the trimer apex. The third panel describes the effect of the removal of N276 glycan on the binding of PGT122 and PGV19 individually, followed by the sequential binding of both antibodies to the N276D mutant. The bottom right illustration shows the distance between the variable regions of PGV19 and PGT122 on adjacent protomers. All values of $K_d$ are nanomolar, enthalpy change ($\Delta H$) is cal/mol, and entropy ($\Delta S$) is cal/mol/deg. Binding stoichiometry ($N$) is directly affected by uncertainties in protein concentration measurement, total active molecules in the sample, and glycan heterogeneity. Associated errors/uncertainties are less than 10% of the average of at least two independent binding measurements

Isothermal titration calorimetry (ITC) reveals that PGT122 and PGV19 bind individually to oligomannose-populated BG505 NFL.664 with $K_d$ values of 699 nM and 806 nM, respectively (Fig. 8, top panel). Recapitulating the sequence of binding events that resulted in the formation of the crystal used for data collection, we observe that binding of PGT122 Fab is enhanced to BG505 NFL.664 after PGV19 is bound ($K_d$ = 346 nM) indicating a positive allosteric modulation of the N332 site after the CD4bs is occupied by PGV19 (Fig. 8, top panel). But interestingly, this modulatory effect is reversed when the trimer is expressed without glycosidase inhibitors in HEK 293F cells. With natural (complex, hybrid, and oligomannose) glycoforms on Env, PGT122 and PGV19 individually bind with $K_d$ values of 345 nM and 217 nM, respectively (Fig. 8, middle panel), while binding of PGV19 to the PGT122-bound BG505 NFL.664 is reduced ($K_d$ = 826 nM). The negative allosteric effect on CD4bs antibody PGV19 after binding of PGT122 to the NFL trimer expressed in HEK 293F cells is clearly affected by the glycosylation state. However, this effect is non-reciprocal and an inversion of the sequence of binding events (i.e., PGV19 followed by PGT122) does not cause as severe a reduction in binding, possibly due to no steric hindrance of the CD4bs-surrounding glycan fence on the N332 bNAb (Fig. 8, middle panel). These results are in agreement with previous observations of allosteric effects on binding antibodies to neighboring Env immunogen epitope clusters[53, 54]. The higher binding affinities observed for trimers produced without glycosidase inhibitors are better estimates of the binding of antibodies in vivo versus trimers having homogeneous oligomannose glycoforms.

We next investigated the role of N276 in accommodation, binding, and allostery of the CD4bs and N332-directed Abs. PGV19 bound with a $K_d$ of 153 nM to the BG505 NFL.664 N276D mutant produced in HEK 293F cells (Fig. 8, bottom panel). As expected, removal of the N276 glycan improves binding (as reflected in the decreased $K_d$ value) of the CD4bs antibody but, counter-intuitively, severely reduces binding of PGT122 ($K_d$ of 741 nM) to the PGV19-bound BG505 NFL N276D mutant. This decreased association ($1/K_d$) is reflected by the slope of the sigmoid curve and difference in the number of data points observed in the saturated state for identical experimental parameters and concentrations of binding partners (Fig. 8, bottom panel), when compared to the original BG505 NFL.664 construct (corresponding antibody binding in Fig. 8, middle panel). Most interesting, however, is that PGV19 binds to the PGT122-bound BG505 NFL.664 N276D mutant with a $K_d$ of 63 nM. This result indicates likely stabilization provided by PGT122 to the N332-supersite glycans when it binds first to the N276D mutant, thus helping PGV19 to bind with higher affinity and minimal heat change. These studies demonstrate that allosteric modulation is related to the glycan composition of the Env, gaps in the glycan fence, and the dynamic nature of glycans rather than the encoded protein, and also determines the direction of allosteric modulation of the immune response for all compact prefusion trimer Env designs that have converging glycosylation profiles.

We also investigated the differences in stability of BG505 NFL.664 trimers containing different glycoforms. The exclusively oligomannose form ($T_m$ = 66.6 °C) has similar stability to that containing natural (oligomannose, complex, and hybrid) glycoforms ($T_m$ = 65.8 °C) (Supplementary Fig. 1a). Binding of PGV19 and PGT122 to HEK 293F cell produced BG505 NFL.664 results in an additional ~3 °C of Env stabilization ($T_m$ = 68.7 °C) (Supplementary Fig. 1b). These thermostability and binding characteristics are similar for the cleavage-independent NFL.664 and cleaved SOSIP.664 BG505 constructs (Supplementary Fig. 2a–c).

## Discussion

The crystal structure of BG505 NFL.664 Env is similar to the well-characterized BG505 SOSIP.664 Env, illustrating that the SOSIP and NFL designs are highly comparable at the structural level. The NFL format recapitulates the presentation of immunogenic epitopes on gp120 and gp41 in context of the Env trimer, as in SOSIP and ΔCT formats of HIV Env, with the added advantage of being furin-cleavage independent. The glycosylation profiles of soluble NFL and SOSIP Env are also similar. The soluble formats have minimal deviations from the oligomannose content found on native viral spikes, although the complex glycans are generally less processed[32, 55]. The flexible linker connecting gp120 and gp41 in a single chain format does not affect either its native-like prefusion compact structure or distort known epitopes on Env. In addition, the longer flexible linker in NFL trimers does not hamper inter-domain flexibility, which is comparable to the furin-cleaved constructs. We also observe a more structurally ordered state of the dynamic HR1_N region and resolve the entire FP without a stabilizing antibody bound at this epitope. We have also determined the quaternary contacts of a VRC01-class lambda antibody PGV19 in the context of a glycosylated trimer and illustrate variations in conserved features among members of this class. Our data indicate that differences in the glycosylation profile (due to different expression systems or addition of glycosylation inhibitors to the same expression system) leads to variation in Env stability and allosteric modulation of the antibodies that engage the engineered Env, which is relevant for HIV vaccine design. An extensive glycan shield was resolved in our crystal structure that indicates many glycans are protected from endoglycosidase activity when antibodies are bound at the CD4bs and at the N332 supersite. This study thus enables a direct comparison of two design formats of the BG505 isolate, where the NFL single-chain format offers an alternative furin-independent platform for both glycoprotein and DNA-based vaccines.

## Methods

**Protein expression and purification.** Soluble HIV Env trimers: Expression and purification of Env NFL trimers were as described previously[20]. Briefly, HEK 293F cells (Life Technologies catalog# R79007,±10 mg swainsonine/2.5 mg kifunensine) were transfected with plasmids encoding soluble BG505 NFL.664 trimer protein (GenScript). The supernatant containing secreted Env was collected and purified by affinity chromatography using *Galanthus nivalis* lectin (GNL) column (Vector Labs). The protein from the trimer peak was pooled and subjected to size exclusion chromatography (SEC) using Superdex200 16/60 or Superdex200 Increase 10/300 GL columns (GE Healthcare Life Sciences), followed by a negative selection step using non-neutralizing mAb F105, and then purified by one more round of SEC (Supplementary Fig. 1). Site-directed mutagenesis (QuickChange Lightning kit, Agilent technologies) was used to generate the N276D mutation in the soluble NFL Env and the trimers were purified as described above.

Antibody purification: PGV19[27], PGT122[54], PGT121[54], and PGT124[23] Fabs (Genscript) were transiently transfected in FreeStyle HEK 293F cells (Invitrogen) and purified by affinity chromatography (human lambda resin from Life Technologies) followed by cation exchange chromatography (GE Healthcare). VRC03[50], VRC01[28], and VRC06[56] Fabs were generated by cleaving the respective IgGs (Genscript) using the Pierce-Fab Preparation Kit (Cat # 44985) and further purified by SEC.

**Thermostability studies.** Differential scanning calorimetry (DSC)**:** The melting temperatures ($T_m$) of the Env NFL trimers and complexes with PGV19 and PGT122 were determined using Microcal VP-Capillary DSC (Malvern) in PBS buffer at a scanning rate of 1 °C/min from 25 °C to 90 °C. The trimers and complexes were analyzed at 0.25 mg/mL. Data were analyzed using the VP-Capillary DSC automated data analysis software (Supplementary Fig. 1).

Differential Scanning Fluorimetry (DSF): DSF measures the change in fluorescence of certain dyes that bind preferentially to the exposed hydrophobic patches of proteins undergoing denaturation/unfolding as a function of temperature[57, 58]. DSF was performed in CFX96 RT-PCR detection system (BIO-RAD, Hercules, CA) using SYPRO orange fluorescent dye (Life Technologies). The thermal stability of BG505 NFL.664 was assessed in the absence or presence of Fabs VRC01, VRC03, and VRC06. In reactions where Fabs were used, Env and Fabs were mixed and incubated for 1 h at 4 °C before addition of the dye. A typical DSF reaction contained 30 μg of HIV Env, ±10 μg Fabs, 6 μL of the 50× dye (diluted in

PBS from the supplied 5000x SYPRO orange stock in DMSO) and the volume was adjusted with PBS to 25 μL in a clear PCR tube. The temperature of the reaction was increased from 20–95 °C at 0.5 °C increments over a period of 1 h 45 min with an equilibration time of 5 s at each temperature prior to measurement. Using an excitation wavelength of 450–490 nm, the emission spectra were collected using a 560–580 nm range fluorescence resonance energy transfer (FRET) filter. The raw data were initially analyzed by CFX Manager (version 1.6) and the sigmoidal dependence of fluorescence as a function of temperature was fitted to Boltzmann equation to calculate the $T_m$ directly from the inflection point of the transition in the sigmoidal fluorescence curve[58].

**Binding analyses**. Biolayer interferometry (BLI): The binding kinetics of VRC01, VRC03 and VRC06 Fabs with BG505 NFL.664 were determined by Octet (FortéBio) using BLI. Histidine-tagged BG505 NFL.664 or the N276D mutant at ~10 μg/mL in binding buffer (PBS) was captured on the surface of anti-His capture biosensors (FortéBio) for 300 s, followed by a 60 s wash in binding buffer to establish a baseline signal. The biosensors were dipped in wells containing either Fab or IgG in serial dilutions (in binding buffer) at initial starting concentrations ranging from 200 to 400 nM. The Env-Fab/IgG association rate (on-rate) and dissociation rate (off-rate) were measured over 120 s intervals.

**Isothermal titration calorimetry**. ITC: ITC experiments were carried out using a MicroCal Auto-iTC2000 (GE) instrument. The BG505 NFL.664, its N276D mutant, and Fabs PGV19 and PGT122 were extensively dialyzed in a buffer composed of 20 mM Tris and 150 mM NaCl (pH 7.4) before being titrated. After dialysis, the protein concentrations were adjusted using calculated extinction coefficients and absorbance at 280 nm. The PGV19 and PGT122 Fabs were treated as ligands and placed in the syringe at concentrations ranging between 80 and 220 μM and the soluble BG505 NFL.664 Env trimers were considered as receptor and placed in the cell at concentrations ranging between 6 and 20 μM. Two-component binding experiments were performed: first, with the Env construct (or its mutant) in the cell and one of the two Fabs in the syringe, followed by the complex formed in the previous experiment in the cell and the remaining Fab in the syringe. The experiments were carried out at 25 °C with 16 injections of 2.5 μL, lasting 5 s each, with an interval time of 180 s, and reference power of 5 μcals. Data were analyzed with a one-site binding model on Origin 7.0 to obtain the dissociation constant ($K_d$), the molar reaction enthalpy ($\Delta H$), and the stoichiometry of binding ($N$).

**Protein complex formation and partial deglycosylation**. Various combinations of CD4 binding site antibodies with BG505 NFL.664 and PGT122 were screened for crystallization after partial deglycosylation with EndoH (New England Biolabs) treatment for 1 h at 37 °C. The Fabs were mixed in a 2:1 molar ratio per binding site on the NFL trimer before further purification of the complexes by SEC (Supplementary Fig. 1).

**Crystallization and data collection**. Partially deglycosylated and SEC-purified BG505 NFL.664 complexes were subjected to extensive crystallization trials with concentrations ranging between 1 and 10 mg/mL, using different crystallization screens set-up using the Oryx crystallization robot (Douglas Instruments) in our lab (4 °C and 20 °C). Crystals were obtained for all complexes screened, but most diffracted in the range of 5–10 Å at synchrotron sources (APS and SSRL). PGV19 Fab was crystallized at ~7 mg/mL in 10% (v/v) glycerol, 0.1 M ammonium sulfate, 0.1 M HEPES (pH 7.5), 5% (w/v) PEG 3000, and 30% (v/v) PEG 400, and diffracted to 2.5 Å resolution at Advanced Photon Source (APS) beamline 23ID-B. The best diffracting crystals, out of all BG505 NFL.664 complexes screened, were grown in 0.1 M HEPES (pH 7.0) and 15% (w/v) PEG4000, and cryo-protected with 25% glycerol. Diffraction data were collected at APS beamline 23ID-D (Supplementary Fig. 1).

**X-ray data processing and structure determination**. X-ray diffraction data processing (integration and scaling) was performed with HKL-2000[59]. The crystal structure of unbound PGV19 was solved by MR using PGV20 (PDB 4LSU) as the search model using Phaser[60]. The BG505 NFL.664+PGV19+PGT122 complex structure was determined by MR with PGV19 (determined in this study) and BG505 SOSIP.664+PGT122 from PDB 4TVP. The structures were refined using Phenix[61] and model building performed in Coot[62]. MolProbity[63], Privateer[64], and pdb-care[65] were used for structure validation. The refinement statistics are reported in Supplementary Table 1.

**Electron microscopy: sample preparation and data collection**. The SEC-purified BG505 NFL.664 trimers were incubated with 10× molar excess of VRC03 Fab for 1 h at room temperature. The trimer-Fab complex was purified by SEC using a Superose 6 column (GE Healthcare) in 50 mM Tris, 150 mM NaCl, pH 7.4. Fractions containing the complex were pooled and concentrated using a 100 kDa cutoff concentrator (Amicon Ultra, Millipore) to 3.5 mg/mL; 5 μL complex was incubated with 1 μL fresh DDM solution at 1.8 mM to obtain thinner ice on the grid. At 4 °C, 3 μL of this mixture was applied to a CF-2/2-4C C-Flat grid (Electron Microscopy Sciences, Protochips, Inc.) that had been plasma cleaned for 5 s using a mixture of Ar/O₂ (Gatan Solarus 950 Plasma system), blotted, and then immediately plunged into liquid ethane using a manual freeze plunger.

Micrographs were collected on a FEI Titan Krios operating at 300 keV coupled with a Gatan K2 direct electron detector via the Leginon interface[66]. Each exposure image was collected at 22,500× nominal magnification resulting in a pixel size of 1.31 Å/pixel in the counting mode, using a dose rate of ~10 e⁻/pix/s, and 200 ms exposure per frame. A total of 3140 micrographs were collected in ~72 h. The total dose in the EM data collection was 32.6 e⁻/Å². The nominal defocus range used was −1.5 to −3.5 μm.

**Electron microscopy data processing**. All of the collected frames were aligned prior to processing[67]. CTF estimation was carried out using CTFFind3[68], and particles were picked using an automated particle-picking program implemented in the Appion software package[69]. Particles were stacked using a box size of 256 × 256 pixels at 1.31 Å/pix in EMAN boxer.py via Appion[70]. Three rounds of reference-free 2D classification were carried out using MSA/MRA[71] with a binning factor of 2, to remove amorphous particles. After 2D classification, the 2× binned particles were subject to 3D classification using RELION[72], starting with an initial reference model of an unliganded trimer filtered to 60 Å resolution without imposing symmetry, and requesting six classes. Two stoichiometric classes were identified in the data: the Env trimer in complex with either two or three copies of VRC03 Fab. Refinements were carried out in RELION with C3 symmetry imposed (three-Fab class) and without symmetry imposed (two-Fab class)[72, 73]. The final resolutions were 7.8 Å (three-Fab containing density map) (Supplementary Fig. 3) and 8.6 Å (two-Fab-containing density map) at an FSC cutoff of 0.143. The FSC was calculated using a soft-edged mask with a Gaussian fall-off, encompassing the entire structure, including the Fab constant regions.

**Mass spectrometry**. Analysis of overall N-glycosylation profiles: N-linked glycans were enzymatically released by in-gel PNGase F (Peptide-N-glycosidase F) digestion from soluble trimers separated by non-reducing SDS-PAGE. The released glycans were subsequently fluorescently labeled using 2-aminobenzoic acid (2-AA) and analyzed by HILIC-UPLC as previously described[11, 74].

Site-specific N-glycosylation analysis by on-line liquid chromatography mass spectrometry: Site-specific N-glycosylation analysis was performed as described[13]. Briefly, tandem ion mobility-ESI MS (ion mobility electrospray mass spectrometry) of the total pool of enzymatically released glycans from Env trimers was performed using a Waters Synapt G2Si instrument (Waters Corp.). These glycan libraries were used for subsequent semi-quantitative site-specific analysis; 200 μg of trimer was subjected to in-solution digestion using trypsin or chymotrypsin (Mass Spectrometry Grade, Promega), respectively. Following glycopeptide enrichment, glycopeptides were analyzed by LC-ESI MS employing a Q-Exactive Orbitrap mass spectrometer (Thermo Fisher) and higher energy collisional dissociation (HCD) fragmentation of glycopeptides. Byonic (version 2.7) and Byologic software (version 2.3; Protein Metrics Inc.) were used for data analysis and glycopeptide identification. The relative abundance of individual glycoforms on specific N-glycosylation sites was determined using the intensities of the extracted-ion chromatograms (XICs) over all charge states.

**MD simulations**. The crystal structure of BG505 NFL.664 was used as a template for MD simulations. The disordered linker region (G⁵⁰⁹.............S⁵¹¹) was modeled using the FREAD modeling server[75]. We built the oligomannose glycans and then attached them to all N-glycosylation sites observed in crystal structure using the Glycam webserver (http://www.glycam.org). The Glycam webserver used the glycam force field for energy minimization of glycans and removed clashes with protein. Under physiological conditions, glycans are normally highly flexible entities and a single static structure cannot represent their dynamic behavior. Thus, we performed MD simulations in explicit water to gain a more complete understanding of the spatial and dynamic properties of this system after addition of hydrogen atoms, counter ions required for electrostatic neutralization of the complex, and solvation box of TIP3P waters[76].

MD simulations in explicit aqueous solvent were performed for glycosylated BG505 NFL.664 with its linker[76–79]. We used the AMBER force field ff99SB for protein[78], while parameters for glycans were taken from the GLYCAM06 force field[79]. The system was solvated in a box of TIP3P water using periodic boundary conditions. First, restrained energy minimization of 10,000 cycles, followed by 20,000 unrestrained minimization cycles was performed to remove unfavorable contacts. The whole system was then slowly heated from 5 to 300 K for 1 ns, followed by an equilibration step of 4 ns at constant temperature (300 K) and pressure (1 atm). Once the system was stabilized, the production phase for the MD simulations was performed for an additional 250 ns under the same conditions using a 2 fs time step. The SHAKE algorithm and the particle-mesh Ewald method were employed during the MD simulations.

**MD trajectory analysis**. The MD trajectory was analyzed with the modules[80] of AmberTools12. The RMSD and RMSF modules were used to analyze each frame of the MD production runs to determine the average overall fluctuation and conformational fluctuation of each residue.

**Data availability**. The coordinates and structure factors reported in this manuscript have been deposited in the Protein Data Bank (PDB) with accession codes 6AVN and 6B0N. The authors declare that all other data supporting the findings of this study are available within the article and its Supplementary Information files, or are available from the authors upon request.

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

## Acknowledgements

We thank Drs. Jean-Philippe Julien, Xiaoping Dai and Xueyong Zhu for helpful discussions. This work was supported by the International AIDS Vaccine Initiative Neutralizing Antibody Center and Collaboration for AIDS Vaccine Discovery (CAVD OPP1084519 to I.A.W., A.B.W., and R.T.W.; OPP1115782 to M.C. and A.B.W.). IAVI's work is made possible by generous support from many donors, including the Bill & Melinda Gates Foundation and USAID. The full list of IAVI donors is available at www.iavi.org. X-ray data sets were collected at the GM/CA@APS-23ID-B and 23ID-D beamlines, which have been funded in whole or in part with Federal funds from the National Cancer Institute (ACB-12002) and the National Institute of General Medical Sciences (AGM-12006). This research used resources of the Advanced Photon Source (APS), a U.S. Department of Energy (DOE) Office of Science User Facility operated for the DOE Office of Science by Argonne National Laboratory under Contract No. DE-AC02-06CH11357. This work was supported by the Scripps Center for HIV/AIDS Vaccine Immunology and Immunogen Discovery (CHAVI-ID) UM1 AI100663 (A.S., S.K., S.K.S., A.I., R.L.S., A.B.W., M.C., R.T. W., I.A.W.), and in part by the Chris Scanlan Memorial Scholarship from Corpus Christi College, Oxford, United Kingdom (A.-J.B.), NIH P01 HIVRAD AI104722 (S.B., R.T.W.), and NIH R56 AI084817 (I.A.W.). The contents of this publication are solely the responsibility of the authors and do not necessarily represent the official views of NIGMS, NIAID, NIH, USAID, or the US Government. The funders had no role in study design, data collection and analysis, decision to publish, or preparation of the manuscript. This is manuscript number 29580 from The Scripps Research Institute.

## Author contributions

Project designed by A.S., S.B., M.C., A.B.W., R.T.W., and I.A.W. Protein expression and purification by A.S. and S.B. X-ray crystallography by A.S.; ITC, BLI, DSC, and DSF by A.S., S.B., and S.K.S. Glycan analyses and mass spectrometry by A.-J.B. and M.C. Electron microscopy by N.d.V., J.P., and A.B.W. DNA amplification and MD simulations by S.K. Data analysis and interpretation by A.S., S.K., S.B., A.I., D.C.D., R.L.S., A.B.W., M.C., R.T. W., and I.A.W. Manuscript written by A.S. and I.A.W. and edited by A.S., S.B., S.K., A.I., R. L.S., A.B.W., R.T.W., M.C., and I.A.W.

## Additional information

**Competing interests:** The authors declare no competing interests.

