## [Peer Review File · Nature Communications]

Reviewers' comments:

Reviewer #1 (Remarks to the Author):

Sarkar et al. reports structural and biochemical analyses of a novel uncleaved SOSIP variant and interactions with a pair of neutralizing antibodies (PGV19 and PGT122). While building on much previous work in these areas, the report extends prior results, and adds enough new material and useful information, to merit publication. The manuscript is clear and well-written, though densely packed with extensive results, not all of which add equally to the overall scientific impact of the manuscript. For instance, it is unclear how the results from the solution thermostability studies add to the interpretive framework. While it is inherently difficult to estimate error from T_m measurements, and I do believe a three degree difference in stability is "significant", I don't know how to interpret a one degree difference, or how these results add to the discussion. In general, experiments are performed, described, and interpreted correctly. The structure of the unbound antibody is well-refined, but the SOSIP/Fab complex is less so, though, in general, it is interpreted appropriately and is informative. One exception is the discussion of hydrogen bonds in the SOSIP complex structure (eg., Supplementary Figure 7), which is not appropriate at this resolution. It would also be nicer to show electron density maps calculated with less biased coefficients than straight 2Fobs-Fcalc. A more significant area of concern are the ITC data, some of which are of variable quality, and which have odd, shifting, derived binding stoichiometries that need to be explained, or at least discussed. And an absolute requirement is to provide error estimates for all the reported binding parameters, ensuring that significant figures are handled properly.

The recommendation is to accept after making the suggested minor revisions. No additional experiments are necessary.

Reviewer #2 (Remarks to the Author):

Sarkar and coworker's manuscript "Crystal structure of a cleavage-independent, soluble HIV Env recapitulates the architecture and glycosylation of the native cleaved trimer" presents the structure of the "NFL" trimer of HIV-1 strain BG505 in complex with CD4 binding site antibody PGV19 and V3-glycan antibody PGT122. NFL trimers differ from the extensively used SOSIP trimers in having stability provided by a non-cleavable flexible linker between gp120 and gp41 rather than an engineered disulfide between gp120 and gp41. Multiple laboratories are developing SOSIP trimers as potential HIV immunogens, and NFL trimers may provide an alternative platform. This study provides additional validation that NFL trimers are native-like.

The manuscript is well written and the structural and biophysical studies appear solid. There are a couple of omissions in the structural comparisons part of the analysis. Most notable is that there is no comparison of the BG505 NFL trimer structure with the 3.9 Å crystal structure of a clade C NFL trimer reported in May 2017 (Immunity 46, 792; rcsb code 5UM8). Also the manuscript does not address whether there any notable differences between the PGV19 binding interactions and that of clonally-related antibody PGV20 in

complex with gp120 (4LSU).

Although this study contributes to the characterization of the NFL trimer format, it does not seem suitable for publication in Nature Communications since the results are mostly anticipated based on previous studies. Verification that the NFL trimer site-specific glycosylation profile closely resembles that of BG505 SOSIP is certainly supportive of further development of NFL immunogens. However, this is not an unexpected result based on the previous NFL trimer crystal structure (along with earlier papers characterizing NFL trimers), which demonstrated that NFL trimers have a stable, native-like closed form. The previous structure was done with NFL trimers stabilized in the prefusion-conformation by having several gp41 helix residues mutated to glycine, which seems not to have been necessary for the BG505 version.

Reviewer #3 (Remarks to the Author):

This a characteristically detailed and interesting study by the Wilson group determining the crystal structure and glycans of NFL trimers and comparing them to SOSIP trimers and other forms of Env that lead the authors to conclude that NFL is comparable to SOSIP as a trimer mimic and reveal a few features that differ from SOSIP. The authors also examine the PGV19 VRC01 class mAb structure and the similarities and some differences with other VRC01 class mAbs as well as allosteric effects on mAb bindings that appear to depend on glycan state. I list below a number of mostly minor comments and suggestions to help clarify/improve the paper below:

The legend to figure 1c and line 162 of the text says the large rmsd between the NFL and deltaCT is due to PGT151 being used for the latter. However, is it also possible that this large rmsd in part derives also partly from bona fide conformational differences between the soluble construct and the membrane one? (as well as a little also from different strains used?). There seem to be several variables, so while PGT151 may well be the biggest effect, the other factors may contribute and its difficult to separate them unambiguously without taking steps to eliminate the variables (which may be difficult). Also, the rmsd's in figure 1b differ a bit from those in the text (line 158). Which is correct?

apo-PGV19 is mentioned on line 113 and 304. I'm personally not familiar with the meaning of "apo-" and so it should probably be described on first mention. I could not find this in the methods either. Perhaps this is an engineered protease cleavage site in the antibody that allows elution of captured trimer from columns? line 304 makes me doubt that, however. It may mean unbound free mAb.. Would be good to clarify somewhere.

N-acetylglucosamine is shortened to GlcNAc (line 75, where it isn't defined) or NAG (line 109 and fig 7a, line 247 etc). It would be good to be consistent.

line 117 was confusing "a substantial part of the glycan shield...". The trimers were treated with EndoH, leaving one GlcNAc "stump", except those protected by the bound mAbs, so the

shield should be largely eliminated? Or, does this mean that endo H treatment was not fully effective because it can not access its target sites, even in places not complexed with MAb? This seems to be the case in later text around line 247? and on line 109 where it says endo H digestion was partial. It was just a bit unsettling to read without that point not being addressed early on.... Perhaps a word or two could be added around line 117 to avoid confusion and to mention its dealt with later? Also, I was confused by the "78 glycan residues on 23 out of 27 N linked sites per monomer". If 23 sites are used, then $23 \times 3 = 69$ glycans per trimer, so I can't see how there can be 78 - unless each GlcNac or Man counts as a residue? This may be a misunderstanding, ...perhaps all this can be clarified? Same thing on line 252, "3.8 glycans per sequon" doesn't have meaning to me. Please clarify.

In Figure 1, residues are not labeled. Features such as the V1V2 and fusion peptide, M530/FPPR referred to in the text around line 120 might be added for orientation, along with a few key residue numbers for non-specialist readers to be able to better decipher the features in the fig that are mentioned in the text.

Line 126: For EM, why were CD4bs antibodies selected rather than others? A few words of explanation might be good. Perhaps because the propellor structure provides a good visual for reconstructions? Why not PGV19 as the melting temp is higher (Supp fig 1)?

Line 422: "natively flexible linker" is a bit of a misnomer. It implies the linker is native, whereas the native really applies to the overall conformation. I think dropping the "natively" is better.

Line 179: Suppl figure 5a. The figure is not labeled a,b,c - there are 5 images with no letters.

Line 184: fusion peptide. Since it's attached to the linker, it's unclear that what we see is the true unbound conformation of a fully native trimer. Like the VRC34 bound SOSIP, it may be structurally different from ground state. It's not mentioned and probably published elsewhere, but do VRC34 FP binders bind well to NFL as compared to SOSIP? It would be good to know that it is capable of adopting the VRC34 conformation seen in SOSIP and that the linker doesn't interfere with this happening. If published elsewhere it could be cited or mentioned briefly.

Line 187: I get that the FP is flanked by the N88 glycan but I don't see the glycan in the image, not sure if that matters? This presumably refers to the Kong paper, where that glycan was present on SOSIP? Although SOSIP-VRC34 is explained in the figure, the text here doesn't clarify that the 'VRC34 bound state' is SOSIP -- so there are two differences (VRC34 binding and SOSIP) that might impact the FP, not just VRC34. Not a big deal, but a word or two might be good to point this out explicitly, as like the similar point above, extra variables may confound an iron clad comparison.

Figure 4. The 5FUU plot seems to show gp140 rmsd's are lower than gp41, but the numbers don't reflect that (both 2.7).

Figure 4c, This comparison here and in Fig 4d and great, although the fig 4c legend/figure should probably mention the R166 residue used to calculate the distances to E654, in line with the text for clarity?

line 262: How does Man2 occur? I'm not aware that trimming or EndoH would lead to this glycan type? Or does this just mean that the rest of the glycan is disordered aside from a Man2 stem? at line 261 it says its ordered then at line 264 largely disordered?

line 284-6/Fig 5c: does the lack of N160 processing impact NFL recognition by V2 apex antibodies that might prefer or require complex glycans? the high mannose microclusters could be specified not just cited and it could be pointed out which ones are not high mannose on unfolded gp120/gp140s. It's my understanding that the high mannose patch is a universal feature that doesn't depend on prefusion state. Though this detail may have been dissected elsewhere, a few words to clarify might be good here regarding what is universal of the high mannose patch and what parts are not. The meaning of the glycans aligned on the x-axis of Figure 5c should be decoded somewhere. Previously published, but should probably be included again so readers can understand - especially for the complex glycan structures.

Line 312 variation says in L1 and H3, but the figure legend suppl fig 6b says L2 and H3. Is it L1 or L2? (or both?). Based on Fig 6c, it appears it should be L1 and H3, so the suppl fig6b might be changed/or L1 differences mentioned here too? The HFR3 could be labeled on Suppl Fig 6b. Line 317 say N280, but it's N179 in the figure. The naive VRC01 class antibody is in figure 6c but not mentioned in the text, maybe it should be?

line 356 talks about PGT122 then PGV19 binding whereas on the inhibitor trimer on the top panel (Figure 8) and bottom not being different and cooperative versus not depending on glycans. but the reverse sequence of addition is disfavored on the 293F trimer and not done on the inhibitor one, so we dont know if there is a steric impact on the inhibitor trimer, calling into question the inference on line 357-358, if I understand correctly (we dont know if it's affected by glycosylation state as the same thing might happen on the kifunenine/swainsonine trimer which wasn't done.) On the other hand, the effects reported for the N276 KO mutant are clear. I'm not sure if the reverse sequence can be helpful if added to the top right and bottom right of this figure to flesh this interesting observation out a bit? also the kds on the control trimers are stronger than on the inhibitor trimers and this isn't specifically commented on, as to what it might imply.

Although the glycosylation profiles of SOSIP and NFL differ from other unfolded forms of Env, is anything known about how well they represent glycan types on viral trimers? Although they are probably more representative than say isolated gp120, how much better are they? Are they near perfect or are there some differences? It might be worth commenting briefly, possibly in the conclusions to put the observations into context with how viral trimers may look and what the comparability and limitations might be.

The conclusions ends nicely, as the potential in DNA vaccines is a possibly big advantage of NFL over SOSIP, as furin cleavage is not required. NFL may be also simpler to manufacture

than SOSIP for similar reasons - although a caveat might be that negative selection is still required to eliminate misfolded trimers and it may be difficult to control this as a (potential) drawback in use as DNA vaccines should these misfolded forms impact neutralizing responses. No need to change any text here, commenting only.

Reviewers' comments:

Reviewer #1 (Remarks to the Author):

Sarkar et al. reports structural and biochemical analyses of a novel uncleaved SOSIP variant and interactions with a pair of neutralizing antibodies (PGV19 and PGT122). While building on much previous work in these areas, the report extends prior results, and adds enough new material and useful information, to merit publication. The manuscript is clear and well-written, though densely packed with extensive results, not all of which add equally to the overall scientific impact of the manuscript.

We thank the reviewer for the helpful suggestions and positive comments on our manuscript.

For instance, it is unclear how the results from the solution thermostability studies add to the interpretive framework. While it is inherently difficult to estimate error from T_m measurements, and I do believe a three degree difference in stability is “significant”, I don't know how to interpret a one degree difference, or how these results add to the discussion.

Response: We thank the reviewer for noticing these details. Comparison of the solution thermostability between the BG505 NFL.664 containing oligomannose versus a variety of glycoforms (oligomannose, hybrid and complex) indeed shows similar thermostability under identical experimental conditions. These results confirm that the soluble HIV Env trimers bearing different glycoforms have comparable melting temperatures (T_m) with neither bearing a significant stability advantage over the other. We have incorporated this interpretation in line 443 of the manuscript.

In general, experiments are performed, described, and interpreted correctly. The structure of the unbound antibody is well-refined, but the SOSIP/Fab complex is less so, though, in general, it is interpreted appropriately and is informative.

Response: We thank the reviewer for these comments. The BG505 NFL.664 trimer in complex with PGV19 and PGT122 has a lower resolution compared to the unbound Fab, but both refined structures deposited in the PDB have good Ramachandran statistics. The higher $R_{\text{free}}/R_{\text{cryst}}$ values for the complex structure might be a result of the anisotropy frequently observed in HIV Env trimer crystals, as well as the lower overall resolution.

One exception is the discussion of hydrogen bonds in the SOSIP complex structure (e.g., Supplementary Figure 7), which is not appropriate at this resolution.

Response: Supplementary Figure 7 describes VRC01-class signature features. This figure compares Fab-gp120 complex structures for VRC01 (resolution 2.68Å), VRC03 (resolution 1.89Å) and PGV19+BG505 NFL.664 trimer complex at (3.39Å). We agree that our trimer complex is at a lower resolution than the other two structures being compared, but being at the combining site the illustrated residues are well-resolved and appear at a reasonable distance to potentially form hydrogen bonds. However, we understand the concern about the resolution of the complex and, hence, have modified the legend of Supplementary Figure 7 to reflect that panels b-e reflect *potential* hydrogen bonds for the NFL complex and also mention the resolution of all complexes being discussed.

It would also be nicer to show electron density maps calculated with less biased coefficients than straight $2F_{\text{obs}}-F_{\text{calc}}$.

Response: We thank the reviewer for this suggestion. Figure 2 and Supplementary Figure 5 have been updated with the 2mFobs-Fcalc composite omit maps to remove model bias.

A more significant area of concern are the ITC data, some of which are of variable quality, and which have odd, shifting, derived binding stoichiometries that need to be explained, or at least discussed. And an absolute requirement is to provide error estimates for all the reported binding parameters, ensuring that significant figures are handled properly.

Response: We thank the reviewer for these observations and comments.

It is not entirely clear to us what the reviewer refers to as odd shifting. Hence we would like to clarify the reasons for some deviations in our data from the ideal ITC plots:

- Top panel – BG505 NFL.664+PGV19 binding: the sharp isolated baseline spike. These typically occur when samples are at temperatures $\geq 15^\circ\text{C}$ lower than the experimental temperature. Buffer exchange for the samples was performed at 4°C , which possibly caused a gas bubble to come out of solution during the binding experiment, which was performed at 25°C , despite allowing time for the sample to reach room temperature.
- The upward spikes in some of the raw binding data are a reflection of instrumental variation.

N values:

ITC N values have been an issue and a point of discussion for experiments performed across labs for HIV Env trimers. For antibodies that bind HIV Env trimers in a 1:1 ratio on 3 binding sites per trimer, a range of N values is reported in literature, for example in (Garces et al., 2015), which has extensive ITC measurements in the paper, reports N values between 1.1 to 3.4 for the N332-supersite binding antibodies. It has been noted by the microcalorimetry instrument manufacturer Malvern that “the ITC N-values are not the same thing as the stoichiometry (binding ratio). N-values do in fact equal the stoichiometry if the concentrations we use for the fitting are correct and 100% active.” The N value will deviate if any or all of the following are true:

* Not 100% of the molecules in the instrument’s cell or syringe during the ITC experiment are active,

Or more technical issues like:

* The nanodrop has an offset in its concentration measurement,

* Total concentration of the samples in the cell or syringe are not accurate

Such issues were also observed with N values in (Julien et al., 2013).

We have modified the legend of Figure 8 to address this concern.

Provide error bars:

Error values are now included in Figure 8 and its figure legend.

The recommendation is to accept after making the suggested minor revisions. No additional experiments are necessary.

Response: We thank the reviewer for the positive recommendation.

Reviewer #2 (Remarks to the Author):

Sarkar and coworker’s manuscript “Crystal structure of a cleavage-independent, soluble HIV Env recapitulates the architecture and glycosylation of the native cleaved trimer”

presents the structure of the “NFL” trimer of HIV-1 strain BG505 in complex with CD4 binding site antibody PGV19 and V3-glycan antibody PGT122. NFL trimers differ from the extensively used SOSIP trimers in having stability provided by a non-cleavable flexible linker between gp120 and gp41 rather than an engineered disulfide between gp120 and gp41. Multiple laboratories are developing SOSIP trimers as potential HIV immunogens, and NFL trimers may provide an alternative platform. This study provides additional validation that NFL trimers are native-like.

The manuscript is well written and the structural and biophysical studies appear solid. There are a couple of omissions in the structural comparisons part of the analysis. Most notable is that there is no comparison of the BG505 NFL trimer structure with the 3.9 Å crystal structure of a clade C NFL trimer reported in May 2017 (Immunity 46, 792; rcsb code 5UM8).

Response: Our study here focuses on the comparison of the BG505 isolate in the NFL and SOSIP formats, and additional SOSIP trimers with comparable stabilizing mutations. However, we agree that this would be a useful comparison. Our published 16055 NFL trimer at 3.9 Å resolution has a protomer/trimer C α RMSD of 0.7Å/0.9Å compared to the 3.39Å BG505 NFL.664 structure, but with a substantially greater number of stabilizing mutations, which was one of the reasons for not having included comparisons to this or other structures with similar higher number of stabilizing mutations in our original submitted manuscript. The 16055 structure has now been included in the section on structural comparison.

Also the manuscript does not address whether there any notable differences between the PGV19 binding interactions and that of clonally-related antibody PGV20 in complex with gp120 (4LSU).

Response: We thank the reviewer for this comment. The only notable difference between PGV19 and PGV20 is at CDRL3 position 91 (PGV19: Tyr91, PGV20: Phe91) and position 58 of the heavy chain (PGV20 has Asn58 like VRC01 while PGV19 has Ser58). We have now incorporated the comparison of VRC01-class signature interactions between the clonally related PGV19 and PGV20 in the text.

Although this study contributes to the characterization of the NFL trimer format, it does not seem suitable for publication in Nature Communications since the results are mostly anticipated based on previous studies. Verification that the NFL trimer site-specific glycosylation profile closely resembles that of BG505 SOSIP is certainly supportive of further development of NFL immunogens. However, this is not an unexpected result based on the previous NFL trimer crystal structure (along with earlier papers characterizing NFL trimers), which demonstrated that NFL trimers have a stable, native-like closed form. The previous structure was done with NFL trimers stabilized in the prefusion-conformation by having several gp41 helix residues mutated to glycine, which seems not to have been necessary for the BG505 version.

Response: We do not agree that our results could be anticipated based on previously published studies. First, at the time we started this study, it was not obvious that the structures in the NFL format would correspond so closely to the structures in the SOSIP format. Second, it was also not clear that the glycan composition in the soluble NFL format would match the glycosylation of the SOSIPs, given the expectation that NFLs would adopt the mature compact form in the endoplasmic reticulum whereas for the SOSIP this would occur at the point of furin cleavage in the Golgi apparatus. Third, the previously published NFL trimer is from a clade C isolate, which in other formats had proven difficult to express as native-like trimers in suitable quantities. Clade C trimers

also have different and additional glycosylation sites as compared to BG505 NFL.664 and results with the 16055 isolate used in the previous study would then differ from this study, especially in C2 (interactions with CD4bs antibodies) and C3 (interactions with N332 antibodies) regions on the Env trimer. Fourth, the previously published clade C NFL structure lacks information on key structural features like the flexible linker, fusion peptide, and HR1N region that we describe in detail in our structure. Fifth, as the reviewer correctly notes, the NFL constructs of the two isolates differ substantially in stability. Thus, it would be presumptuous to infer similarity in behavior and molecular interactions. Sixth, the way in which glycoforms affect interactions and implications of their removal on the allostery between epitopes on the soluble Env immunogens have not been reported before in structural detail, based on complementary results from x-ray crystallography, molecular dynamics simulation and electron microscopy, and also the quantification of inter-domain flexibility and subdomain rotation. Our study provides a full characterization of BG505 (presently the most well-characterized HIV Env isolate) that has been designed as both the NFL and SOSIP Env immunogens being tested in animal models. Hence, characterizing the NFL BG505 Env trimer is critical as it enables a direct comparison between these two candidate HIV vaccine immunogens and provides insights for guiding structure-based improvements in immunogen designs to target key epitopes such as the receptor-binding site and the fusion peptide that are surrounded by the glycan shield. Hence, we definitely believe for the reasons above that this paper is worthy of publication in Nature Communications.

Reviewer #3 (Remarks to the Author):

This a characteristically detailed and interesting study by the Wilson group determining the crystal structure and glycans of NFL trimers and comparing them to SOSIP trimers and other forms of Env that lead the authors to conclude that NFL is comparable to SOSIP as a trimer mimic and reveal a few features that differ from SOSIP. The authors also examine the PGV19 VRC01 class mAb structure and the similarities and some differences with other VRC01 class mAbs as well as allosteric effects on mAb bindings that appear to depend on glycan state. I list below a number of mostly minor comments and suggestions to help clarify/improve the paper below:

The legend to figure 1c and line 162 of the text says the large rmsd between the NFL and deltaCT is due to PGT151 being used for the latter. However, is it also possible that this large rmsd in part derives also partly from bona fide conformational differences between the soluble construct and the membrane one? (as well as a little also from different strains used?). There seem to be several variables, so while PGT151 may well be the biggest effect, the other factors may contribute and its difficult to separate them unambiguously without taking steps to eliminate the variables (which may be difficult). Also, the rmsd's in figure 1b differ a bit from those in the text (line 158). Which is correct?

Response: We thank the reviewer for helpful suggestions to improve our manuscript. We attribute the large root-mean-square deviation (RMSD) between BG505 NFL.664, described here, and the published cryo-EM JRFL Δ CT structure, on observations that PGT151 induces asymmetry in the trimer. This is supported by our Supplementary Fig. 4c, where the individual protomers of JRFL SOSIP versus Δ CT structures are quite similar, with no apparent major conformational differences, but deviate in the context of the trimer when PGT151 is bound. We thank the reviewer for noticing the difference in RMSDs between the text and Fig 1b. We have now updated the values in the figure.

apo-PGV19 is mentioned on line 113 and 304. I'm personally not familiar with the meaning of "apo-" and so it should probably be described on first mention. I could not find this in the methods either. Perhaps this is an engineered protease cleavage site in the antibody that allows elution of captured trimer from columns? line 304 makes me doubt that, however. It may mean unbound free mAb.. Would be good to clarify somewhere.

Response: We thank the reviewer for this comment and we understand that it might confuse readers who are not familiar with this terminology in structural immunology. Apo- indicates an unbound form of a protein, in this context of the antibody, as the reviewer correctly surmises later in the comment. We have now replaced "apo" with "unbound" throughout the manuscript to avoid such confusion.

N-acetylglucosamine is shortened to GlcNAc (line 75, where it isn't defined) or NAG (line 109 and fig 7a, line 247 etc). It would be good to be consistent.

Response: We thank the reviewer for noticing this point. We have now included a footnote on first mention at line 86, indicating that N-acetylglucosamine is abbreviated as GlcNAc and/or NAG.

line 117 was confusing "a substantial part of the glycan shield...". The trimers were treated with EndoH, leaving one GlcNAc "stump", except those protected by the bound mAbs, so the shield should be largely eliminated? Or, does this mean that endo H treatment was not fully effective because it can not access its target sites, even in places not complexed with MAb? This seems to be the case in later text around line 247? and on line 109 where it says endo H digestion was partial. It was just a bit unsettling to read without that point not being addressed early on.... Perhaps a word or two could be added around line 117 to avoid confusion and to mention its dealt with later? Also, I was confused by the "78 glycan residues on 23 out of 27 N linked sites per monomer". If 23 sites are used, then $23 \times 3 = 69$ glycans per trimer, so I can't see how there can be 78 - unless each GlcNAc or Man counts as a residue? This may be a misunderstanding, ...perhaps all this can be clarified? Same thing on line 252, "3.8 glycans per sequon" doesn't have meaning to me. Please clarify.

Response: We would like to clarify that endoH is an enzyme that cleaves between the two GlcNAc residues in the chitobiose core of the oligomannose or hybrid N-glycan. We have now made word changes in the text on line 123 to make this clearer. On line 132 by "... a substantial part of the N-glycan shield ..." we imply that a large part of the Env glycan shield observed in our electron density remains unaffected by EndoH activity, which is now explicitly mentioned on line 132. This is due to the combined effect of inaccessibility of the glycans when they are parts of glycan microclusters as well as protection from binding antibodies, as the reviewer correctly observes also from line 279. The CD4 binding site antibodies have a larger effective footprint as demonstrated in (Lyumkis et al., 2013) that prevents accessibility to the EndoH enzyme, which we now refer to in the text. On line 133, "78 glycan residues" refers to 78 saccharide moieties (text now modified to make the sentence clearer). We observe these on 23 out of the total 27 N-glycosylation sites present in the BG505 NFL construct. In (Stewart-Jones et al., 2016), a value of 3.8 saccharides per sequon is reported without EndoH treatment, which is close to the 3.4 saccharides per sequon that we observe on our BG505 NFL.664 structure after EndoH treatment. This implies that the specific antibodies bound on our structure largely protect the glycan shield against EndoH activity. Here below is the

explanation of what glycans per sequon means. We have provided a better explanation in the text on lines 133-134 and 287-288.

The value of average number of saccharide moieties per sequon was calculated as follows:

$78 \text{ saccharide moieties} \div 23 \text{ N-glycosylation sites for which we had clear density (0.8 to } 1.0 \sigma) = 3.4 \text{ saccharide moieties per N-glycosylation site that was observed.}$

In Figure 1, residues are not labeled. Features such as the V1V2 and fusion peptide, M530/FPPR referred to in the text around line 120 might be added for orientation, along with a few key residue numbers for non-specialist readers to be able to better decipher the features in the fig that are mentioned in the text.

Response: We have now labeled specific residues in Figure 1, and for orientation, labeled some additional regions of Env in Figure 2.

Line 126: For EM, why were CD4bs antibodies selected rather than others? A few words of explanation might be good. Perhaps because the propeller structure provides a good visual for reconstructions? Why not PGV19 as the melting temp is higher (Supp fig 1)?

Response: Indeed, as the reviewer notes, this was done to help identify compact trimers, which present a pronounced effect on its propeller structure when bound to CD4bs antibodies, which could be compared to the PGV04 cryo-EM structure in complex with BG505 SOSIP for comparison between the two immunogen design formats. We did not use PGV19 despite it providing greater stability to the Env as we already had obtained crystals that diffracted for PGV19 and provided an antibody that bound to the CDbs for the crystal structure determination.

Line 422: "natively flexible linker" is a bit of a misnomer. It implies the linker is native, whereas the native really applies to the overall conformation. I think dropping the "natively" is better.

Response: We thank the reviewer for pointing this out. We have now modified the text to "flexible linker" throughout the text.

Line 179: Suppl figure 5a. The figure is not labeled a,b,c - there are 5 images with no letters.

Response: We thank the reviewer for noticing this point. Supplementary figure 5 has now been modified with updated labels.

Line 184: fusion peptide. Since it's attached to the linker, it's unclear that what we see is the true unbound conformation of a fully native trimer. Like the VRC34 bound SOSIP, it may be structurally different from ground state. It's not mentioned and probably published elsewhere, but do VRC34 FP binders bind well to NFL as compared to SOSIP? It would be good to know that it is capable of adopting the VRC34 conformation seen in SOSIP and that the linker doesn't interfere with this happening. If published elsewhere it could be cited or mentioned briefly.

Response: We thank the reviewer for this comment. We see the conformation of the entire fusion peptide in the NFL (single chain) format without the stabilizing effect from VRC34.01 as it is not present in our structure. A part of the fusion peptide on our BG505 NFL.664 structure matches that of the fusion peptide on BG505 SOSIP bound to VRC34.01 (PDB 5I8H) as shown in the top inset of Figure 2c. However, the

unbound region on our structure deviates from the SOSIP+VRC34.01 conformation as mentioned in the text. Fusion peptide antibodies binding to the NFL constructs have not been published previously. Comparisons between the fusion peptide antibodies binding to NFL versus SOSIP are part of another manuscript (in preparation). We have therefore made modifications to this (fusion peptide) part of the text (lines 209 to 223) to clarify that the “unbound form” refers to the fusion peptide on BG505 NFL.664 construct when it is not bound to any fusion peptide antibody as well as commented on implications when fusion peptide antibodies bind to NFL constructs.

Line 187: I get that the FP is flanked by the N88 glycan but I don't see the glycan in the image, not sure if that matters? This presumably refers to the Kong paper, where that glycan was present on SOSIP? Although SOSIP-VRC34 is explained in the figure, the text here doesn't clarify that the 'VRC34 bound state' is SOSIP -- so there are two differences (VRC34 binding and SOSIP) that might impact the FP, not just VRC34. Not a big deal, but a word or two might be good to point this out explicitly, as like the similar point above, extra variables may confound an iron clad comparison.

Response: We thank the reviewer for this comment. Indeed, in (Kong et al., 2016), N88 is described as part of the VRC34.01 epitope. In our structure, we observe one GlcNAc residue on N88 that is now shown in the updated Figure 2c. As also mentioned in the response to the previous comment, we have now made changes in the text to specify that the VRC34.01-bound state is on the SOSIP which partly resembles the conformation of the fusion peptide observed on our BG505 NFL.664 structure that does not have any fusion peptide antibody bound. The fusion peptide appears to be quite flexible from variations observed in different structures and the location where the fusion peptide resides (up or down conformations).

Figure 4. The 5FUU plot seems to show gp140 rmsd's are lower than gp41, but the numbers dont reflect that (both 2.7).

Response: We thank the reviewer for this comment. We have now updated the plot on Figure 4a.

Figure 4c. This comparison here and in Fig 4d and great, although the fig 4c legend/figure should probably mention the R166 residue used to calculate the distances to E654, in line with the text for clarity?

Response: We think that the phrase that we used in the text may have been confusing. The distances at the base of the trimer were calculated using C α coordinates of E654 on each of the three gp41 domains of the trimer as mentioned in the legend for Figure 4c. We have now rephrased this description (lines 264-265) in the main text and in the legend for Figure 4d.

line 262: How does Man₂ occur? I'm not aware that trimming or EndoH would lead to this glycan type? Or does this just mean that the rest of the glycan is disordered aside from a Man₂ stem? at line 261 it says its ordered then at line 264 largely disordered?

Response: We have added a sentence on line 296 and rephrased the text in this section to clarify our point. Man₂ refers only to the observed density that is visible for Man₂GlcNAc₂ in our structure and is not a glycoform resulting from EndoH treatment.

line 284-6/Fig 5c: does the lack of N160 processing impact NFL recognition by V2

apex antibodies that might prefer or require complex glycans? the high mannose microclusters could be specified not just cited and it could be pointed out which ones are not high mannose on unfolded gp120/gp140s. It's my understanding that the high mannose patch is a universal feature that doesn't depend on prefusion state. Though this detail may have been dissected elsewhere, a few words to clarify might be good here regarding what is universal of the high mannose patch and what parts are not. The meaning of the glycans aligned on the x-axis of Figure 5c should be decoded somewhere. Previously published, but should probably be included again so readers can understand - especially for the complex glycan structures.

Response: All NFL constructs of Env that are neutralization sensitive to apex antibodies are recognized by PGT145 and PGDM1400 as previously published in (Guenaga et al., 2015; Guenaga et al., 2016; Sharma et al., 2015). N160 lies at the trimer apex and may have slight variation in its terminal glycan processing, possibly due to trimer breathing, between different batches of transiently produced soluble HIV envelopes. The oligomannose patch is a distinguishing feature of well-folded trimers in contrast to uncleaved pseudotrimers and clusters of gp120 as reported in (Behrens et al., 2017). N160 is one of the seven glycosylation sites that form the trimer-associated mannose patch on the BG505 isolate that shows minor populations of mixed or complex glycans between batches of transiently produced soluble Envs. Thus, we observe (lines 317-322) that purely oligomannose populations were found on N160 in this particular analysis of BG505 NFL.664. We have now mentioned the individual glycans on the BG505 isolate known to be a part of the trimer-associated mannose patch as per the reviewer's suggestion and also commented on the variation within the minor populations observed in well-folded native-like compact soluble trimers based on our results. We have also added a description of the glycans on the x-axis of Figure 5c in the figure legend.

Line 312 variation says in L1 and H3, but the figure legend suppl fig 6b says L2 and H3. Is it L1 or L2? (or both?). Based on Fig 6c, it appears it should be L1 and H3, so the suppl fig6b might be changed/or L1 differences mentioned here too? The HFR3 could be labeled on Suppl Fig 6b. Line 317 say N280, but it's N179 in the figure. The naive VRC01 class antibody is in figure 6c but not mentioned in the text, maybe it should be?

Response: We thank the reviewer for these comments. We have now modified the text (line 359) and the figure legend to reflect the observation that there is variation in CDR length and/or conformation in L1, L2 and H3 between PGV19 and VRC03. We have also labeled the regions discussed in this section of the text on Supplementary Fig 6b. We have modified the text on line 365 to reflect the interaction between N279_{gp120} and Trp on CDRH3 as shown in Supplementary Fig 7c. We have now incorporated additional information about the VRC01-class naive antibody HuGL2 in this section of the text to coincide with the description of the CDR sequences shown in Fig 6c.

line 356 talks about PGT122 then PGV19 binding whereas on the inhibitor trimer on the top panel (Figure 8) and bottom not being different and cooperative versus not depending on glycans. but the reverse sequence of addition is disfavored on the 293F trimer and not done on the inhibitor one, so we dont know if there is a steric impact on the inhibitor trimer, calling into question the inference on line 357-358, if I understand correctly (we dont know if it's affected by glycosylation state as the same thing might happen on the kifunenine/swainsonine trimer which wasn't done.) On the

other hand, the effects reported for the N276 KO mutant are clear. I'm not sure if the reverse sequence can be helpful if added to the top right and bottom right of this figure to flesh this interesting observation out a bit? also the kds on the control trimers are stronger than on the inhibitor trimers and this isn't specifically commented on, as to what it might imply.

Response: We thank the reviewer for discussing this aspect of the binding studies and we apologize at the outset for the lengthy response. All BG505 NFL.664 trimers were produced in HEK 293F cells; those used for crystallography were produced with kifunensine+swainsonine to have only oligomannose glycoforms (highly pure and homogenous samples have better chances of crystallization); biophysical studies were performed using soluble Env trimers produced in both HEK 293F with (Fig 8, top panel) and without (Fig 8, middle and lower panels) glycosidase inhibitors kifunensine and swainsonine.

Fig 8 (top panel) replicates the sequence of antibodies binding to BG505 NFL.664 to form the complex resulting in the diffracting crystal. On solving the crystal structure, with both PGT122 and PGV19 bound, we found the variable regions of the Fabs binding on these two separate epitopes on adjacent trimers close enough to impact binding. On close inspection we found N301 to be directly in the way of CD4bs antibodies (Fig 8 bottom illustration); N301 contains complex glycans (Cao et al., 2017; Go et al., 2011; Gristick et al., 2016) when soluble trimers are produced in 293F cells without glycosidase inhibitors.

- Since the soluble trimers used for crystallization only contained oligomannose glycoforms, we did not proceed with testing the reverse sequence for these soluble Env trimers, as it would not be relevant to the glycan composition of viruses. Instead, we performed binding experiments with those trimers produced without the glycosidase inhibitors. The steric impact on the HEK 293F cell produced NFL+PG19+PGT122 is clear on comparison with the same experiment with the trimers expressed with glycosidase inhibitors. Unlike at the CD4bs, antibodies targeting the N332 supersite had no major steric challenge to overcome and our results of improved binding of PGT122 to the PGV19-bound BG505 NFL.664 containing oligomannose glycoforms indicate that this particular glycoform does not create any hindrance in binding of the N332-directed antibodies. Also, similar binding observations were reported by (Julien et al., 2013), with PGT121 family of antibodies and another clade A isolate (KNH1144), but in the absence of a high resolution structure at the time, a clear reason for this observation could not be identified.
- We agree with the reviewer that doing a reverse sequence of binding experiments with the trimers produced without glycosidase inhibitors (NFL.N276D+PGT122+PGV19) would provide completion to the binding results and thus have now added this panel to Fig 8, lower panel and discuss the implications on lines 433-437.
- Indeed, K_d 's of antibodies binding to trimers produced with glycosidase inhibitors are greater than those observed for trimers produced without inhibitors. This is possibly a closer reflection of antibody binding *in vivo* with the soluble Env trimer having a virus-like glycan composition during immunogen production (without glycosidase inhibitors). We have now added this comment to lines 420-422.

Although the glycosylation profiles of SOSIP and NFL differ from other unfolded

forms of Env, is anything known about how well they represent glycan types on viral trimers? Although they are probably more representative than say isolated gp120, how much better are they? Are they near perfect or are there some differences? It might be worth commenting briefly, possibly in the conclusions to put the observations into context with how viral trimers may look and what the comparability and limitations might be.

Response: The reviewer is correct in noting that the soluble forms of the HIV Env trimers differ from the viral spike and gp120s alone. Results from (Pritchard, Harvey, Bonomelli, Crispin, & Doores, 2015) identify the mannose content on the soluble Envs to be largely similar to viral spikes with slight elevation in the oligomannose content in the soluble forms (possibly due to restrictions on flexibility due to stabilizing mutations). However, the recombinant trimer was found to have higher levels of complex glycans than the virion (Panico et al., 2016), although the complex glycans on the peripheral blood mononuclear cells-derived virions and pseudoviruses were found to be more processed (bi-, tri, tetra-antennary N-glycans) compared to the recombinantly produced soluble Env which contained smaller/less-processed complex glycans (Pritchard et al., 2015). We now briefly note these possible deviations from the virus glycosylation patterns in the conclusion.

The conclusions ends nicely, as the potential in DNA vaccines is a possibly big advantage of NFL over SOSIP, as furin cleavage is not required. NFL may be also simpler to manufacture than SOSIP for similar reasons - although a caveat might be that negative selection is still required to eliminate misfolded trimers and it may be difficult to control this as a (potential) drawback in use as DNA vaccines should these misfolded forms impact neutralizing responses. No need to change any text here, commenting only.

Response: We thank the reviewer for the positive comments. We would like to clarify that although negative selection is a protocol originally started for purification of HIV NFL constructs, purification by positive selection (PGT145 and 2G12) has also been used successfully to purify NFL trimers to yields similar to SOSIPs (unpublished data). However, we take the point that some misfolded trimers may be produced in a DNA vaccine format but we do not know yet whether this will affect or skew the immune response.

References:

- Behrens, A. J., Harvey, D. J., Milne, E., Cupo, A., Kumar, A., Zitzmann, N., . . . Crispin, M. (2017). Molecular architecture of the cleavage-dependent mannose patch on a soluble HIV-1 envelope glycoprotein trimer. *J Virol*, *91*, e01894-16. doi:10.1128/JVI.01894-16
- Cao, L., Diedrich, J. K., Kulp, D. W., Pauthner, M., He, L., Park, S. R., . . . Paulson, J. C. (2017). Global site-specific N-glycosylation analysis of HIV envelope glycoprotein. *Nat Commun*, *8*, 14954. doi:10.1038/ncomms14954
- Garces, F., Lee, J. H., de Val, N., de la Pena, A. T., Kong, L., Puchades, C., . . . Wilson, I. A. (2015). Affinity maturation of a potent family of HIV antibodies is primarily focused on accommodating or avoiding glycans. *Immunity*, *43*, 1053-63. doi:10.1016/j.immuni.2015.11.007

- Go, E. P., Hewawasam, G., Liao, H. X., Chen, H., Ping, L. H., Anderson, J. A., . . . Desaire, H. (2011). Characterization of glycosylation profiles of HIV-1 transmitted/founder envelopes by mass spectrometry. *J Virol*, *85*, 8270-84. doi:10.1128/JVI.05053-11
- Gristick, H. B., von Boehmer, L., West, A. P., Jr., Schamber, M., Gazumyan, A., Golijanin, J., . . . Bjorkman, P. J. (2016). Natively glycosylated HIV-1 Env structure reveals new mode for antibody recognition of the CD4-binding site. *Nat Struct Mol Biol*, *23*, 906-15. doi:10.1038/nsmb.3291
- Guenaga, J., Dubrovskaya, V., de Val, N., Sharma, S. K., Carrette, B., Ward, A. B., & Wyatt, R. T. (2015). Structure-guided redesign increases the propensity of HIV Env to generate highly stable soluble trimers. *J Virol*, *90*, 2806-17. doi:10.1128/JVI.02652-15
- Guenaga, J., Garces, F., de Val, N., Stanfield, R. L., Dubrovskaya, V., Higgins, B., . . . Wyatt, R. T. (2016). Glycine substitution at helix-to-coil transitions facilitates the structural determination of a stabilized subtype C HIV envelope glycoprotein. *Immunity*, *46*, 792-803.e793. doi:10.1016/j.immuni.2017.04.014
- Julien, J. P., Sok, D., Khayat, R., Lee, J. H., Doores, K. J., Walker, L. M., . . . Wilson, I. A. (2013). Broadly neutralizing antibody PGT121 allosterically modulates CD4 binding via recognition of the HIV-1 gp120 V3 base and multiple surrounding glycans. *PLoS Pathog*, *9*, e1003342. doi:10.1371/journal.ppat.1003342
- Kong, R., Xu, K., Zhou, T., Acharya, P., Lemmin, T., Liu, K., . . . Mascola, J. R. (2016). Fusion peptide of HIV-1 as a site of vulnerability to neutralizing antibody. *Science*, *352*, 828-33. doi:10.1126/science.aae0474
- Lyumkis, D., Julien, J. P., de Val, N., Cupo, A., Potter, C. S., Klasse, P. J., . . . Ward, A. B. (2013). Cryo-EM structure of a fully glycosylated soluble cleaved HIV-1 envelope trimer. *Science*, *342*, 1484-90. doi:10.1126/science.1245627
- Panico, M., Bouché, L., Binet, D., O'Connor, M.-J., Rahman, D., Pang, P.-C., . . . Morris, H. R. (2016). Mapping the complete glycoproteome of virion-derived HIV-1 gp120 provides insights into broadly neutralizing antibody binding. *Sci Rep*, *6*, 32956. doi:10.1038/srep32956
- Pritchard, L. K., Harvey, D. J., Bonomelli, C., Crispin, M., & Doores, K. J. (2015). Cell- and protein-directed glycosylation of native cleaved HIV-1 envelope. *J Virol*, *89*, 8932-44. doi:10.1128/JVI.01190-15
- Sharma, S. K., de Val, N., Bale, S., Guenaga, J., Tran, K., Feng, Y., . . . Wyatt, R. T. (2015). Cleavage-independent HIV-1 Env trimers engineered as soluble native spike mimetics for vaccine design. *Cell Rep*, *11*, 539-50. doi:10.1016/j.celrep.2015.03.047
- Stewart-Jones, G. B., Soto, C., Lemmin, T., Chuang, G. Y., Druz, A., Kong, R., . . . Kwong, P. D. (2016). Trimeric HIV-1-Env structures define glycan shields from clades A, B, and G. *Cell*, *165*, 813-26. doi:10.1016/j.cell.2016.04.010

REVIEWERS' COMMENTS:

Reviewer #1 (Remarks to the Author):

The revised manuscript addresses all concerns adequately, and is now acceptable for publication.

Reviewers' comments:

Reviewer #1 (Remarks to the Author):

The revised manuscript addresses all concerns adequately, and is now acceptable for publication.

Response: We thank the reviewer for the positive recommendation.